# Metal Toxicity and Dementia Including Frontotemporal Dementia: Current State of Knowledge

**DOI:** 10.3390/antiox13080938

**Published:** 2024-08-01

**Authors:** Francesca Gorini, Alessandro Tonacci

**Affiliations:** Institute of Clinical Physiology, National Research Council, 56124 Pisa, Italy; francesca-gorini@cnr.it

**Keywords:** artificial intelligence, environmental pollution, frontotemporal dementia, FTD, toxic metals, machine learning, neurodegeneration, neurotoxicity, oxidative stress, toxicity

## Abstract

Frontotemporal dementia (FTD) includes a number of neurodegenerative diseases, often with early onset (before 65 years old), characterized by progressive, irreversible deficits in behavioral, linguistic, and executive functions, which are often difficult to diagnose due to their similar phenotypic characteristics to other dementias and psychiatric disorders. The genetic contribution is of utmost importance, although environmental risk factors also play a role in its pathophysiology. In fact, some metals are known to produce free radicals, which, accumulating in the brain over time, can induce oxidative stress, inflammation, and protein misfolding, all of these being key features of FTD and similar conditions. Therefore, the present review aims to summarize the current evidence about the environmental contribution to FTD―mainly dealing with toxic metal exposure―since the identification of such potential environmental risk factors can lead to its early diagnosis and the promotion of policies and interventions. This would allow us, by reducing exposure to these pollutants, to potentially affect society at large in a positive manner, decreasing the burden of FTD and similar conditions on affected individuals and society overall. Future perspectives, including the application of Artificial Intelligence principles to the field, with related evidence found so far, are also introduced.

## 1. Introduction

Frontotemporal dementia (FTD) represents a neurodegenerative condition accounting for 10% of middle-aged-onset dementias, and representing the third overall cause of dementia across all ages, after Alzheimer’s disease (AD) and Lewy body dementia (LBD). It imposes a significant burden from social and economic perspectives, as well as impacting regional and national healthcare systems, on both patients and their caregivers, and features important clinical signs, including behavioral abnormalities, speech disturbances, and neuropsychiatric symptoms. Until the early 2010s, FTD was extremely underdiagnosed, mainly due to its partial clinical overlap with characteristic features of AD, but it was later characterized by an incidence of 2.36 cases per 100,000 persons per year in Europe [1,2], with an average age at diagnosis of 56 years. In some cases, diagnoses were reported at as early as 20 years of age, although these were often misclassified as other psychiatric disorders, including schizophrenia [3]. The main contribution to the development of FTD is represented by genetic causes, accounting for 30–50% of cases, depending on the studies; however, non-genetic risk factors are also present, including neuroinflammatory processes, the presence of autoimmune disorders, a history of head trauma, specific language disorders in early age, or right-hemispheric dominance patterns [4]. More specifically, in the neuroinflammatory framework, it has been well ascertained that toxic metals, including―albeit not limited to―lead (Pb) and cadmium (Cd), can impair synaptic transmission and foster neuroinflammation, therefore impacting the health status of the central nervous system (CNS) [5]. However, studies in this specific scenario are quite few and heterogeneous, leading to inconclusive or scattered evidence, mainly due to the scarce clinical characterization of patients with FTD, their misdiagnoses until a few years ago, as previously reported, the difficulty of retrieving personal histories of metal exposure, as also occurred with other clinically relevant cohorts, as well as the poor consideration of new tools and technologies for conducting correlation studies going beyond the traditional statistical analysis commonly adopted in such a framework.

Under such premises, the current review seeks to overcome such limitations and aims to review the actual literature evidence about the specific topic, where it exists, and to derive similarities from studies dealing with clinical conditions that are similar to FTD in terms of etiopathological and physiopathological features. This will enable us to draw possible hypotheses about this relationship, paving the way for observing possible future outlooks in terms of the use of new technologies and approaches derived from Artificial Intelligence (AI) that could be useful in retrieving existing correlations between clinical, epidemiological, and instrumental variables, which are currently latent according to the traditional statistics.

### 1.1. Tauopathies: General Features

The term “tauopathy” refers to a group of neurodegenerative disorders with heterogeneous biochemical, morphological, and clinical signs, although they all share the symptoms of cognitive decline and dementia [6]. These diseases are characterized by the pathological accumulation of a family of intrinsically disordered proteins, the highly soluble microtubule-associated protein tau (MAPT), which are localized in the CNS and expressed especially in neurons, although they may also be detected in glial cells and extracellular space (e.g., interstitial fluid and cerebrospinal fluid) [7,8,9,10]. More than 26 different tauopathies are currently known, encompassing a wide range of phenotypically diverse diseases that, along with FTD, also include, among others, AD, argyrophilic grain disease (AGD), corticobasal degeneration (CBD), chronic traumatic encephalopathy (CTE), Parkinson’s disease (PD), Pick’s disease (PiD), and progressive supranuclear palsy (PSP) [8,10,11]. In humans, tau proteins are encoded by a single gene, MAPT, located on chromosome 17q21 and comprising 16 exons, whose expression, following the alternative messenger RNA (mRNA) splicing of exons 2, 3, and 10, gives rise to six distinct isoforms [9,11]. In particular, the presence or absence of exon 10 in the microtubule-binding compartments leads to the generation of six isoforms with four repeated microtubule-binding domains (4R) or with three microtubule-binding domains (3R) [10]. Thus, tauopathies can be classified into 3R tauopathies (mainly containing 3R tau), 4R tauopathies (with a main prevalence of 4R tau), and 3R/4R tauopathies (having an almost equal distribution between 3R and 4R tau contents; see Table 1) [9,10,12]. Interestingly, the presence of an extra copy of exon 4 generates a high-molecular-weight isoform, known as “big tau”, which has been speculated to improve axonal transport and, due to the paucity of phosphorylated sites and their increased size, to have a low propensity for aggregation [13]. Physiologically, tau resides in neuron axons, where it promotes the assembly and stabilization of microtubules, contributing to regulating axonal transport and neurite growth, and in the nucleus, where it binds directly to adenine–thymine-rich α-satellite heterochromatin, regulating DNA packaging [11,14]. If tau normally undergoes post-translational modifications (phosphorylation, acetylation, methylation, ubiquitination, truncation), the hyperphosphorylation of serine and threonine residues in tau isoforms results in the disruption of tau–microtubule interaction and the consequent instability of the cytoskeleton, and also enhances the ability of tau to self-aggregate and accumulate in intracellular insoluble neurofibrillary tangles (NFTs), as was recently shown in vitro [9,15,16,17]. Furthermore, based on the so-called “prion paradigm”, experimental studies using cell cultures and animal models have documented that tau aggregates exhibit prion-like seeding properties; therefore, tau may proliferate as a misfolded protein according to a “nucleation elongation” process and propagate systematically along connected neurons through both synaptic contacts and non-synaptic mechanisms [11,18]. Tau propagation can ultimately result in the impairment of axonal transport, neuronal dysfunction, and neurodegeneration, eventually leading to cell death, although the propagation of tau aggregates appears to be independent of toxicity, showing some features that do not reflect the “prion-like” model [19,20].

So far, 60 mutations in the MAPT gene, most of them being missense mutations, have been linked to the onset of neurodegenerative diseases [7,9,10]. Primary tauopathies, belonging to the heterogeneous group of FTDs comprising MAPT-associated FTD, AGD, CBD, CTE, PiD, and PSP, and characterized by behavior, language, and motor impairment, exhibit as their hallmark deposits of aggregated tau proteins in neurons and glial cells in the cerebral cortex and in other brain regions [11,21,22]. Most cases of primary tauopathies are sporadic, and only around 30% of patients have a family history of the disease. MAPT missense mutations functionally affect tau, while those that alter mRNA splicing are capable of both impairing microtubule assembly and increasing the amount of the free, unbound form of tau in the cytosol, thus inducing tau aggregation (reviewed studies in [10,11]). Notably, there is currently no clear correlation between MAPT mutations and post-translational modifications of tau [7], although mutated tau proteins appear to be more susceptible substrates for hyperphosphorylation than their wild-type counterparts [23]. As for secondary tauopathies, which include AD, LBD, PD, Down syndrome, Huntington’s disease, myotonic dystrophy, and Niemann–Pick disease type C, no pathogenetic mutations can be detected in the MAPT gene, nor does tau play any primary pathogenic role, and the development of tau pathology occurs in response to other pathogenic events (e.g., the formation of amyloid beta-Aβ, the main component of neuritic plaques in AD) [6,24].

### 1.2. Frontotemporal Dementia: Etiopathogenesis, Clinical Signs, and Epidemiology

FTD, having as a major trait the degeneration of the frontal and/or temporal lobes of the brain with typical severe neuronal loss, gliosis, and spongiosis of the superficial layers, is a highly heritable condition; 30–50% of FTD patients have a positive family history of dementia [22,25,26]. In particular, the heritability of FTD is primarily characterized by autosomal dominant transmission (detected in 10% of familial disease) in three main genes, i.e., MAPT, progranulin, and chromosome 9 open reading frame 72 (C9orf72), with each of them accounting for 5–10% of overall FTD cases [2,27,28]. Recently, other genes have been found to be associated with autosomal dominant FTD, although they are only responsible for less than 5% of all FTD cases [28]. Despite their overall pathological heterogeneity, approximately 40% of all FTD cases are caused by tau pathology, with a widespread deposition of tau in neurons and glial cells [17]. Based on linkage analyses conducted in the 1990s, patients with mutations in the MAPT gene and tau deposits in neuronal and glial cells were classified as having FTD and parkinsonism linked to chromosome 17q21 (FTDP-17) [21]. Afterwards, the identification of many kindreds with familial FTDP-17 presenting a clinical phenotype indistinguishable from that of cases with known MAPT mutations, but without MAPT mutation or tau pathology, led to the discovery of different types of mutations and other causative genes [21,29]. Currently, the term FTD encompasses various clinical manifestations based on the dominant symptoms at diagnosis, including the behavioral variant of FTD (bvFTD), the semantic variant, the nonfluent variant of primary progressive aphasia, the right-lobe variant, and FTD associated with motor neuron disease (FTD-MND), which have as their main features the impairment in interpersonal skills and behavior, speech and semantic memory, personality, and motor neurons, respectively, with bvFTD being the most frequent variant in all genetic groups [2,21,27,28,30] (Figure 1).

As stated above, all the mutations in MAPT, with a few exceptions, are transmitted through a dominant inheritance pattern with complete penetrance [21]. Furthermore, in FTD and other tauopathies, such as CED, PD, and PSP, the MAPT gene presents two haplotypes: while the H1 haplotype can be detected in all ethnic groups, the H2 haplotype has been identified in European and southwest Asian populations and has been associated with a reduced risk of neurodegenerative diseases [6].

Together with other disorders such as AD, FTD causes dementia, a syndrome characterized by a deterioration in cognitive function that is frequently accompanied or preceded by changes in emotional status, mood, and behavior, and whose incidence generally increases with age, being more common in subjects aged 65 years and older [31].

In general, according to the WHO, more than 55 million people live with dementia globally, with over 60% of them residing in low- to middle-income countries, where access to care is often scattered. Aging is considered to be the most important related risk factor, and every year, 10 million people are diagnosed with dementia worldwide, with global costs amounting to USD 1.3 trillion, mainly devoted to informal care provisions. Furthermore, dementia ranks seventh among the major causes of death, in addition to representing one of the main risk factors of disability among older adults at the global level [31].

In addition, so-called “early-onset dementia” (EOD) develops before the age of 65 and includes primarily AD (EOAD), followed by FTD (EOFTD), vascular dementia, and alcohol-related dementia, as well as other conditions [32]. According to a recent Italian population-based study, the incidence of EOD was 6.5/100,000 inhabitants per year after adjustment for sex and age, and the prevalence was equal to 74.3 cases per 100,000 inhabitants, considering the age range 30–64 years.

As previously stated, in the broad spectrum of dementia, AD is the most common form, representing around 60–70% of all cases. Other subtypes comprising the big picture include vascular dementia and LBD, as well as diseases contributing to FTD. Focusing on FTD, it was estimated to account for 2.6% of all dementia cases from a global perspective. In general terms, although it is considered to be the most common form of dementia in presenile age, its prevalence peak occurs between 75 and 79 years of age [33,34], with the age-adjusted incidence of the disorder summing up to 2.90 cases per 100,000 persons per year. FTD patients with initial motor symptoms have the shortest survival time with respect to other variants of the pathology, whereas survival time is quite similar between bvFTD and AD [33]. Actually, most cases of FTD are sporadic and, with increasing age, are possibly the result of a complex interaction between genetic and environmental determinants [22,35,36]. Hence, the multifactorial inheritance pattern involves, on the one hand, multiple loci capable of modulating the individual susceptibility to the disease and, on the other, environmental factors that may trigger or protect against altered endogenous mechanisms [22]. In this scenario, the evaluation of the exposome, defined as the set of both external stimuli (biological and chemical agents, radiation) and internal chemical sources throughout life, together with the analysis of the impact of lifestyle and socio-economic status on exposure and directly on health, has proven to be a relevant approach to elucidate the role of environmental determinants in multifactorial diseases [35,37]. Importantly, early exposure to risk factors during pre- and post-natal periods could predispose people to the onset of neurodegenerative diseases later in life [38]. The association of exposure to certain metals/metalloids, i.e., Cd, Pb, aluminum (Al), arsenic (As), and mercury (Hg), with the increased risk of the development of neurodegenerative disorders has gained importance, given the persistence of such compounds in the environment and their high toxicity at multiple levels to human health, and, therefore, opens a challenge to provide further, decisive insights into the etiopathogenesis of dementia and, in particular, of FTD.

## 2. Toxic Metals and Neurodegenerative Disorders

Since 1 June 2020, As, Cd, Hg, and Pb have been considered among the 10 most harmful chemicals to human health by the World Health Organization, which provided risk management recommendations to protect humans from their harmful effects [39]. Consistent with this, in 2022, the U.S. Agency for Toxic Substances and Disease Registry has included As, Pb, Hg, and Cd within the top 10 substances showing the most significant toxic (known or suspected) effects in humans [40]. In fact, in recent decades, there has been an increase in contamination by toxic metals due to their growing applications in mining, manufacturing, farming, industrial discharge, and technology [41,42]. In addition to anthropogenic activities, these elements are naturally introduced into the environment thanks to events such as volcanic eruptions and rock weathering [43]. Exposure to toxic metals occurs through diverse modes, including ingestion through diet, which represents the most relevant route, inhalation, and, occasionally, dermal absorption, while parameters such as speciation, dose, and duration of exposure can influence the effects, and the consequent toxicity, of these elements in humans [42,44]. Physico-chemical soil properties, such as pH, redox potential, soil texture, organic matter content, and the presence of iron and manganese oxides and ions in the soil solution, may influence the bioavailability of metals and their capacity to transfer and accumulate in both edible and inedible parts of plants and, therefore, enter food chains [45,46]. Being non-essential elements for living organisms and not subject to biodegradation, often with a long half-life, they can accumulate in the human body, causing harmful health effects [43]. Importantly, they may cross the blood–brain barrier (BBB) and accumulate in the brain or, alternatively, enter the CNS via the olfactory pathway [47,48] (Figure 2).

Overall, these toxic elements share common mechanisms, including the production of free radicals (reactive oxygen species—ROS and reactive nitrogen species), and bind to sulfhydryl groups, resulting in the inactivation of crucial molecules in cells, the depletion of reduced glutathione (GSH), the inhibition of antioxidant defense, which in turn can lead to compromise of metabolic pathways, the alteration of hormone homeostasis, the dysfunction of organ and immune systems, and even the development of cancer [49].

The “toxic metal hypothesis” has recently been proposed as a potential explanation for both the detection of toxic metals in human astrocytes, neurons, and oligodendrocytes, as well as the variety of clinicopathological features documented in patients with common neurodegenerative disorders, which would result from different combinations of exposures to toxic metals and underlying genetic variants [50]. Actually, given the essential role of certain trace elements, such as copper (Cu), manganese (Mn), iron (Fe), and zinc (Zn), in the brain, the metal hypothesis links the neuropathology of AD and other neurodegenerative disorders to their homeostatic dysfunction [5]. While experimental studies on animal models have allowed us to evaluate several issues, e.g., the levels of toxicants’ uptake by the CNS, which cells contain the toxicants, and how the toxicants overcome the BBB and the placenta, studies using autopsies performed on brains from subjects that died from neurodegenerative disorders provide a more valuable assessment of metal distributions in these diseases, which generally occur in later ages and are difficult to analyze in young animals [50]. On the other hand, people with neurodegenerative diseases can live for many years after the onset of symptoms, and in the meantime, there may be a loss of cells containing toxic metals [50]. The toxic metal hypothesis entails two possible pathways (see [50] for details):
The selective uptake of toxicants from locus coeruleus neurons, which results in a reduced secretion of norepinephrine from these cells, with consequent damage of the BBB, allowing the transit of metals towards astrocytes;The intact BBB makes possible a slow accumulation of toxic metals in astrocytes, leading to astrocyte dysfunction and metal transfer to neurons and oligodendrocytes.

In the following sections, we will discuss the current evidence on the potential role of Al, As, Cd, Hg, and Pb as triggers for the onset of FTD, also referring, more generally, to the risk of dementia, while considering that FTD and AD, the most common diseases causing dementia syndrome, have different genetic backgrounds, molecular drivers, affected regions of the CNS and, at least in part, different clinical symptoms.

### 2.1. Aluminum

Al, which is ubiquitous and is the third most abundant element in the Earth’s crust, may combine with more than 270 different minerals [51]. Due to its unique chemical and physical properties (i.e., low weight and density, high electrical and thermal conductivity, elevated malleability), Al is largely used in metallurgy and construction and the electric power, pharmaceutical, food packaging and processing, and cosmetics industries, thereby leading to widespread contamination in all environmental matrices [51,52]. Human exposure occurs via food consumption, which is responsible for around 95% of the total daily intake of Al, drinking water, cosmetics (e.g., Al chlorohydrate in antiperspirants and deodorants), and medicinal products like antacids [51,53]. Various forms of Al salts (e.g., Al phosphate, Al hydroxide, Al potassium sulfate, or amorphous Al hydroxyphosphate sulfate) have also served, and are currently used, as adjuvants in vaccines to enhance the effectiveness of vaccination, although a recent meta-analysis and Trial Sequential Analysis, based on 10 randomized clinical trials of healthy volunteers, found neither significant benefits nor risks in the use of Al adjuvants, depending on the concentration, number of doses, or particle size [54]. The average human intake of Al in the general population, as assessed in several European countries, has been estimated to be between 14 and 105 mg per week for a 70 kg adult [55], an amount that may exceed the recommendations of the European Food Safety Authority (EFSA), which in 2008 established a tolerable weekly intake (TWI) of 1 mg Al/kg of body weight (bw) per week [56]. Among food items, cereals and cereal products, vegetables, beverages, and certain infant formulae represent the main contributors to the dietary Al intake [56]. Upon food ingestion, Al is absorbed through the intestinal mucosa and then almost totally binds to the human serum iron carrier transferrin, which distributes Al in all tissues, including the brain through the BBB and the choroid plexuses in the cerebrospinal fluid, and it mainly accumulates in bones, where it reaches levels between 5 and 10 mg/kg [56]. Al can also reach the placenta and fetus and can be detected in breast milk [56]. The absorbed Al, which is less than 1% of the ingested metal [57], is effectively eliminated through the urine as citrate salt, while the unabsorbed portion is excreted via the feces and bile [56]. Despite its widespread diffusion in the environment, Al has no biological role, and Al exposure, in addition to neurotoxicity, has been associated with a variety of health effects, such as impairment of lung function and pulmonary lesions [52,58], cancers [59,60], dyslipidemia, obesity [61,62], and cardiovascular disease [63].

#### 2.1.1. Aluminum and Dementia: The Epidemiological Evidence

So far, only one human study has evaluated the risk association between Al exposure and the occurrence of FTD, but a relatively larger number of investigations have explored the influence of this metal on the risk of dementia. The Italian case–control study by Adani et al. [35], which included 58 patients with EOD (19 EOFTD and 32 EOAD), showed that Al exposure in the work setting was associated with an increased risk of EOD, although the association was not significant and imprecise and almost entirely driven by EOFTD (odds ratio—OR = 4,1 with 95% confidence interval—95% CI: 0.5–34.5). Notably, although it was based on a very limited number of subjects, which resulted in a low statistical power of estimates, this study documented that the excess was primarily influenced by EOFTD cases, probably due to a high prevalence of subjects with FTD-MND, an FTD variant displaying genetic, clinical, and neuropathological overlap with amyotrophic lateral sclerosis [35,64]. Within a systematic review aimed at identifying potential risk factors for dementia, out of the 16 studies included (14 cross-sectional and two prospective cohort studies) investigating exposure to Al in relation to dementia and including a total of around 22,000 subjects, 7 of them reported a positive association between Al concentration in drinking water and increased risk of dementia [65]. The largest study included was a prospective cohort population-based study examining the association between individual exposure to Al in drinking water and risk of dementia and AD in 1677 subjects aged 65 and older in France, which estimated a relative risk (RR) of 1.28 (95% CI: 1.05–1.58, *p* = 0.017) and a RR = 1.34 (95% CI: 1.09–1.65, *p* < 0.006) per increase of 0.1 mg/day of Al in drinking water for dementia and AD, respectively [66]. Despite the relevance of these findings, the authors emphasized that only exposure via drinking water was assessed, although foods, which may contribute to 95% of the total daily Al intake, provide approximately 25 times more Al to the systemic circulation and potential Al body burden, compared to drinking water [66,67]. The systematic review by Killin et al. [65] also summarized the existing evidence on the relationship between occupational exposure to Al and dementia (a total of four studies, of which three were cross-sectional and one used a retrospective cohort), documenting conflicting results, with positive, inverse, and no association. In particular, a positive correlation was found in an Australian cohort study performed on 1894 male underground gold miners, 60 of whom suffered from AD [68]. An increased mortality rate, although not statistically significant compared to the general population (standard mortality rate = 1.38, 95% CI: 0.69–2.75), and a non-significantly higher mortality risk (hazard ratio—HR = 2.76, 95% CI: 0.88–8,82) were observed among miners who inhaled Al dust [68]. However, the authors did not exclude the possibility that these findings likely underestimated the risk due to the underreporting of AD on death certificates and the limited International Classification of Diseases (ICD) coding available for AD [68]. A systematic review and meta-analysis including three cohort and five case–control studies (some of them included in [65]) including a total of 10,567 individuals and evaluating chronic exposure to Al both in the occupational setting and in drinking water (Al concentration equal to or above 100 µg/L) estimated an OR of 1.71 (95% CI: 1.35–2.18) related to the risk of AD, with no significant heterogeneity between studies and excluding the possibility that this association could be attributed to publication bias [69]. Furthermore, the pooled OR comparing subjects consuming drinking water with an Al content exceeding 100 µg/L and those who did not was equal to 1.95 (95% CI: 1.47–2.59) [69]. A subsequent systematic review aimed at providing an overview of the association between metal levels in biological samples and the main neurodegenerative disorders included 12 case–control studies dealing with the relationship between Al and the risk of AD [70]. In that review, four studies reported significantly higher serum Al concentrations in subjects with AD, five studies reported lower levels of the metal in serum, hair, and bone in AD cases compared to controls, and inconclusive results regarding differences between AD patients and controls in terms of Al concentrations in blood, hair, and CSF were highlighted in three investigations [70]. If overall Al levels appear to potentially be involved in the etiology of AD, the heterogeneity of biological specimens between studies imposes a need for caution in interpreting results [70]. In a prior cross-sectional study conducted in northern Italy on 64 workers formerly exposed to Al dust and 32 unexposed controls from other companies, measured serum Al levels in retired smelter workers were nearly twice as high as those of the control group, and Al levels remained high up for to 10 years following exposure [71]. Exposed subjects had worse scores on the mini-mental state examination (MMSE), which is widely used as a screening test for dementia [72], and the serum Al concentration significantly and negatively affected the scores of the MMSE and other neuropsychological tests after adjusting for potential confounding factors (age, body mass index, education, smoking habits, alcohol and coffee intake), suggesting a central role of Al in early neurotoxic effects detectable at a preclinical stage [71]. A systematic review and meta-analysis including 17 studies comparing circulatory Al concentrations in 1295 participants (483 AD patients and 812 controls) documented significantly higher Al levels in AD cases than in control subjects (standardized mean difference—SMD = 1.08, 95% CI: 0.66–1.50, *p* < 0.001), although a significant heterogeneity and publication bias were found across the studies [73]. An Indian case–control study also showed that subjects with AD had significantly higher blood Al levels compared to an identical number (*n* = 50) of age-matched controls [74]. Conversely, the authors did not detect any significant correlation between Al imbalance and upregulation of target genes in AD patients, probably due to the small number of subjects enrolled [74]. A recent case–control study by Babić Leko et al. [48] was conducted on 193 Croatian participants (124 with AD, 50 with mild cognitive impairment, and 19 healthy controls) in order to explore the association of AD using selected AD biomarkers in cerebrospinal fluid (CSF) with macro- and microelements measured in CSF and plasma in the three groups of subjects using both simple correlation and machine learning methods. The authors found that CSF levels of Al positively correlated with levels of phosphorylated tau isoforms, visinin-like protein 1 (VILIP-1), a neuronal calcium sensor protein, pregnancy-associated plasma protein A (PAPP-A), a marker of oxidative stress [75], and albumin, which may augment the risk of AD by inducing Aβ accumulation [76], thus providing relevant insights into the role of toxic metals in AD risk [48].

In summary, despite the near lack of studies focused on FTD, research investigating the risk of AD as it relates to Al exposure has provided moderate evidence of an association, reporting a significantly higher serum concentration of Al in subjects with AD compared to non-demented controls. However, most published studies were based on a retrospective case–control design and current exposure, while symptoms of dementia can appear for several years following exposure. In addition to using larger sample sizes, future longitudinal studies should consider all possible routes of past exposure, including those linked to dietary habits, drinking water, work environments, and the use of certain personal care products; they should use hair or nails as biospecimens, and possibly also evaluate multiple exposures to other toxic elements beyond Al. The study of Al’s influence on specific gene expressions in AD could represent a further relevant element to establish a causal relationship between Al exposure and the onset of AD (Table 2).

#### 2.1.2. Aluminum Neurotoxicity: The Underlying Mechanisms

Although the exact process of the neurotoxicity of Al is still far from being understood at the molecular level, a number of possible mechanisms have been proposed to support the potential role of Al in dementia pathogenesis (reviewed in [77,78]):Exposure to high doses of Al may induce an accumulation of hyperphosphorylated tau in the cortical neurons [79]. Indeed, Al may both inhibit the dephosphorylation of tau and promote its non-enzymatic phosphorylation, causing it to subsequently bind to phosphorylated amino acids, inducing the aggregation of this protein into insoluble deposits and the formation of NFTs, a common feature of all tauopathies [80,81]. Notably, chronic exposure to Al causes greater progressive tau aggregation, apoptosis, and neurological dysfunction in transgenic mice with a pre-existing pathological process resulting in tau accumulation [82].Al and Aβ peptides, which are generated by the proteolysis of the Aβ precursor protein (APP), colocalize in amyloid fibers in the cores of senile plaques in the hippocampal and cortical tissues of the human brain, which is the pathological hallmark of AD [83,84]. Al also has a direct inhibitory effect on the lysosomal cathepsin B-mediated proteolysis of Aβ peptides, thereby enhancing the deposit of neuritic plaques [85]. A diet enriched in Al may also increase Aβ levels and accelerate plaque deposition in the transgenic mouse Tg 2576, which is considered a potential model of AD [86]. Conversely, Gómez et al. [87] did not observe any relevant differences in the genotype between Tg 2576 and wild-type animals exposed to Al for 6 months; thus, their findings did not support the existence of an etiological role of Al in the onset of AD.Due to its oxidation status (+3), Al can strongly bind to oxygen-donor ligands, deprotonated hydroxyl groups, carboxylate, and inorganic and organic phosphates like the phosphate groups of DNA and RNA, influencing the expression of genes involved in crucial roles for brain functioning, including cerebral proteases and monoamine oxidase isotypes; the latter flavin-containing enzymes are located on the outer membrane of mitochondria and are associated, at high levels, with increasing age and AD development [80,88].Al is also able to inhibit the activity of enzymatic antioxidants such as glutathione peroxidase (GPx), superoxide dismutase (SOD), and catalase (CAT), replacing metals essential to redox balance, and to bind to metals like Fe and Cu, which are directly implicated in metal-based oxidative events [89]. In fact, through the formation of stable Al–superoxide radical complexes, Al promotes the Fenton reaction, a catalytic process that initially oxidizes Fe^2+^ to Fe^3+^ in the presence of H_2_O_2_, forming a radical hydroxyl and, in the subsequent step, along with the reduction of Fe^3+^, it generates the hydroperoxide radical [77,89]. Therefore, Al triggers oxidative stress by increasing the concentration of ferrous and ferric ions in the brain, which leads to the impairment of numerous signaling cascades, Fe-mediated lipid peroxidation, tissue damage, and, ultimately, cell apoptosis [78,80,89].Al may also cause the production of highly reactive oxy and hydroxyl free radicals through direct effects on mitochondria, the main site of cellular energy generation and oxygen consumption, resulting in increased production of ROS, which can promote oxidative damage to mitochondrial DNA (mtDNA) and lead to cell death [90]. Importantly, impaired mitochondrial dynamics have been considered as a preclinical marker in neurodegenerative diseases [91].Al appears to be responsible for the increased upregulation of pro-inflammatory or pro-apoptotic signaling elements, including APP, nuclear factor kappa-light-chain-enhancer of activated B cells (NF-κB) subunits, interleukin (IL)-1β precursor, cytosolic phospholipase A2, cyclooxygenase-2, and DAXX, a regulatory protein known to induce apoptosis and repress transcription [92]. Conversely, the metal induces a concentration-dependent decrease in the expression of neurotrophins, including nerve growth factor and brain-derived neurotrophic factor [93].Al exhibits high toxicity to the cholinergic system by altering the synthesis and metabolism of acetylcholine and disrupting cholinergic transmission [94]. In particular, the metal appears to enhance the activity of the enzyme acetylcholinesterase (AChE), which is responsible for the hydrolysis and the consequent inactivation of acetylcholine [95,96]. Furthermore, the bond of Al to the anionic site of the enzyme can trigger the formation of free radicals, which further promotes the observed dysfunction of AChE [80].Finally, Al may interfere with calcium ion (Ca^2+^) homeostasis at multiple levels. In fact, Al induces the release of Ca^2+^ from intracellular stores, determining a peak of Ca^2+^ levels in neuronal cytoplasm and a consequent reduced Ca^2+^ influx, which in turn inhibit phosphoinositide signaling pathways and cause the removal of Ca^2+^ from the cytoplasm to occur at a slower rate [80]. Additionally, the interaction of Al with adenosine triphosphate (ATP) can enhance the action of neurotransmitters [97]. Indeed, neurons and glia release ATP during glutamatergic neurotransmission, and both glutamate and ATP induce an inward movement of Ca^2+^ through N-methyl-D-aspartate (NMDA) receptor channels following their simultaneous bond to post-synaptic receptors [97]. The complex Al-ATP reduces the sensitivity of Ca-dependent inactivation of NMDA receptors, which remain open for longer, causing a persistent elevation of intracellular Ca^2+^ concentration and a consequent excitotoxicity induced by neuronal apoptosis [97]. Al also decreases Ca^2+^ uptake through voltage-gated calcium channels (VGCCs), which play a key role in coupling electrical activity to neurotransmission in a pH- and concentration-dependent manner [98].

### 2.2. Arsenic

As is a ubiquitous metalloid representing one of the most abundant elements in the Earth’s crust and among the major risk factors for public health [49,99]. As is used in electronics and industrial manufacturing, wood preservatives, and, to a limited extent, in pesticides and pharmaceuticals, while natural sources of As include volcanic activity, weathering of rocks, geothermal waters, and forest fires [99,100]. It exists in metalloid, inorganic, and organic forms, and as arsine. The inorganic species arsenate (As^V^) and arsenite (As^III^) are considered to be more hazardous compared to elemental As and the organic arsenical monomethylarsonic acid (MMA^V^) and dimethylarsinic acid (DMA^V^), while the trivalent species monomethylarsinous acid (MMA^III^) and dimethylarsinous acid (DMA^III^) exhibit a greater toxicity than As^III^ due to their higher affinity for sulfhydryl groups [100,101]. The greatest threat to human health arises from abnormal concentrations of As in groundwater, which primarily results from erosion and leaching of geological formations [99,102]. It has been recently estimated that up to 220 million people, the vast majority of them residing in Asian countries (e.g., Bangladesh, India, China), are potentially exposed to As concentrations of 100 μg/L or greater in groundwater, which are much higher than the currently recommended provisional limit of 10 μg/L [102,103]. Exposure to As can also occur through inhalation in agriculture and industrial settings and ingestion of food items, with fish, shellfish, meat, poultry, dairy products, and cereals containing appreciable concentrations of As [100,102], while rice, rice-based products, grains, and grain-based products (excluding rice) are the main contributors to dietary exposure to inorganic As (iAs) [104]. Following ingestion, As is rapidly and in large proportions absorbed in the small intestine (45–80%) and widely distributed to organs and tissues, including several brain regions, owing to the ability of As to cross the BBB [49,104,105]. The cellular biomethylation of iAs to MMA^V^ and DMA^V^ by alternating the reduction of As^V^ to As^III^ serves as a detoxification process; thus, after a half-life of approximately 4 days, As is excreted in the urine in its methylated forms, among which DMA^V^ is the most abundant (40–80% of the total content) [100,101,106]. The International Agency for Research on Cancer (IARC) has listed As and As compounds, as well as iAs in drinking water, among carcinogens to humans [107]. Indeed, long-term exposure to As may cause cancer of the skin, bladder, and lungs, along with other effects like developmental effects, diabetes, pulmonary disease, cardiovascular disease, and adverse pregnancy outcomes [102]. In the update of its 2009 risk assessment on As in food, the EFSA Panel on Contaminants in the Food Chain (CONTAM) concluded that a reference point of 0.06 μg iAs/kg bw per day should be established for skin cancer, and that is within the range of 0.03–0.15 μg iAs/kg bw per day (range of the estimates of mean dietary exposure to iAs in adults). This limit should also protect against lung and bladder cancers, skin lesions, ischemic heart disease, chronic kidney disease, respiratory disease, adverse reproductive effects, and neurodevelopmental disorders [104].

#### 2.2.1. Arsenic and Dementia: The Epidemiological Evidence

Although evidence from experimental studies (described in the following section) supports the neurotoxicity of As, epidemiological studies specifically aimed at evaluating the association of As exposure with the risk of FTD are currently lacking. On the other hand, following the formulation of the “arsenic exposure hypothesis”, advanced by Gong and O’Bryant [108], a set of studies evaluated the potential of As exposure to increase the risk for AD in humans. In 2011, a community-based study involving 434 American rural-dwelling adult and elderly participants showed that both current and long-term low-level groundwater As exposures were significantly associated with poorer scores in language, visuospatial skills, and executive functioning [109]. However, it is important to note that, although the mean and median levels of As in groundwater estimated using geographic information system-based methods were below the current acceptable standard of 10 µg/L, and long-term low-level As exposure impaired a greater number of roe in terms of their neuropsychological functions than those associated with the current exposure, the lack of direct measurements of As in groundwater and participant biospecimens represent two relevant limitations of this study [109]. The systematic review by Killin et al. [65] included two studies exploring the relationship between As exposure and AD. The first showed that slight variations in soil As concentration were related to increased rates of morbidity and mortality due to AD and other dementias in eleven European countries, although one of the major limitations of this study was the lack of information on individual data and those linked to the other routes of exposure such as inhalation and ingestion by food and drinking water [110]. The second study, comparing indicators for selected health outcomes in the Spring Valley of Washington, which was characterized by As contamination in surface soil, found that annual average age-adjusted mortality rates from AD were not statistically different from those in a control area [111]. However, the ecological study design did not allow researchers to evaluate individual-level exposures and outcomes [111]. A more recent ecological study performed in 22 provinces and three municipal districts in mainland China reported a significant correlation between soil As concentration and AD mortality in the general population [46]. Furthermore, the quartile-risk analysis showed an apparent soil As concentration-dependent increase in AD mortality, providing further evidence for the involvement of As in the etiology of AD [46]. In 2010, Baum and co-authors [112] measured the levels of 12 metals in the sera of 44 AD and 41 control subjects, and, although there were no significant differences in serum As concentration between the two groups, they observed a positive and strong correlation of As with MMSE score among individuals with AD. This finding is probably due to the reduced risk of AD associated with the intake of n-3 fatty acids, in particular docosahexaenoic acid, which seafood is rich in [112,113]. Furthermore, while a high intake of iAs, mainly found in drinking water, can compromise cognitive function, seafood As is generally represented by non-toxic organic species [112]. A Korean case–control study testing 89 patients with AD and 118 cognitively healthy individuals for trace metals in sera detected no difference in As levels between the two groups, and As was not significantly correlated with scores in cognitive function [114]. Similar results were reported by Yadav et al. [74], who documented the absence of As in the sera of 50 patients mainly belonging to the Indian population who had been diagnosed with AD. Within a hospital-based case–control study (Lithuanian participants grouped into 53 AD cases and 212 subjects free from AD and dementia), Strumylaite and co-authors [115] observed no significant association between a continuous increase in total urinary As and risk of AD after adjusting for confounding factors. In contrast, a Chinese hospital-based case–control study including 170 patients with AD and 264 controls, after stratifying urinary levels of As species into quartiles, reported that participants’ AD risk increased significantly with higher urinary iAs% and MMA%, whereas it decreased with higher DMA% [116]. The authors further revealed that by combining iAs%, MMA%, or DMA% and selenium (Se), the risk for AD of subjects with a low median level of Se and a high median level of iAs%, or/and a low median level of DMA%, had an approximately 2–3-times-higher AD risk (OR = 2.88; 95% CI: 1.42–5.84 and OR = 2.33; 95% CI: 1.13–4.81, respectively, after adjusting for confounders) [116]. In addition, subjects with higher urinary levels of iAs had worse scores on the MMSE and orientation and recall tests, whereas higher urinary levels of DMA were associated with higher scores on the MMSE and recall tests [116]. Overall, if these findings indicate that a urinary As profile can be associated with an increase in AD risk, the measurement of As from a single urine sample may not reflect past exposure; therefore, repeated measurements over time would be recommended to obtain more reliable estimates. Furthermore, the authors also showed that the overall severity of AD in the participants was mild; therefore, a selection bias may have occurred. More recently, a Turkish case–control study carried out on 40 patients with AD at different stages of severity and 40 healthy control subjects reported that As levels in hair and nail specimens were significantly higher in the case group than in healthy controls (*p* < 0.001) [117]. Importantly, nails contain all forms of iAS, but not DMA^III^, and they primarily reflect the cumulative As exposure from groundwater, but the authors admitted that other intrinsic factors like metabolic or genetic mechanisms of AD could influence this relationship [117]. The study by Babić Leko et al. [48], previously mentioned, showed that As concentrations measured in plasma were significantly and positively associated with levels of certain CSF biomarkers for AD, i.e., VILIP-1 and neurofilament light chain, with the latter being predictive of axonal damage in neurological disorders [118].

In addition to the absolute lack of research on the relationship between As and FTD, to date, the role of As in the development of dementia has been poorly investigated. The main relevant findings on the association of environmental As exposure with morbidity or mortality regarding AD have been shown in descriptive and community-based studies, which, as they are subject to the ecological fallacy, do not allow inferences to be made from the aggregate to the individual levels and are often unable to adjust for important confounders (life habits, genetics setting). Conversely, the analytical studies have provided conflicting results, especially when they were based on As measurements in urinary samples, which, similarly to As contents in serum specimens, reflect only recent exposure, although AD is a chronic disorder. Therefore, large longitudinal cohort studies should be conducted differentiating As species in biospecimens such as nails to assess both past exposures and the link between As metabolism capability and risk of dementia, and information should be collected on a wide range of confounding factors, particularly regarding genetic analysis, diet, and potential sources of exposure (Table 3).

#### 2.2.2. Arsenic Neurotoxicity: The Underlying Mechanisms

As is a well-known environmental toxicant, and, although a growing body of evidence has reported that even low concentrations of As impair neurological and cognitive functions [119], the exact mechanism of action of As remains to be elucidated (see [105,120,121,122] for more details). However, the following impacts of As exposure have been described:As may induce oxidative stress via several pathways: (i) the oxidation of As^III^ to As^V^ results in the generation of H_2_O_2_, and the depletion of GSH as well, thereby hampering the metabolic processing of iAs and exposing the organism to a reduced defense capacity against ROS-mediated damage; (ii) As indirectly promotes ROS generation by decreasing the activities of antioxidant enzymes such as CAT, SOD, and glutathione reductase (GR); (iii) As may also act on the mitochondrial electron transport chain, which is the main responsible mechanism for cell ROS production. Indeed, As inhibits succinic dehydrogenase activity and promotes the uncoupling of oxidative phosphorylation in the ETC, thus inducing an excessive production of superoxide anion radicals that, reacting with nitric oxide, give rise to highly reactive peroxynitrites; (iv) DMA^III^ may specifically induce ROS generation in the endoplasmic reticulum (ER), probably by binding to sulfhydryl groups of proteins in the ER, which suppress the formation of disulfide bonds in newly produced proteins [123].As plays a critical role in mitophagy, which, in the presence of moderate ROS production, is a type of autophagy indispensable for mitochondrial quality control by removing aged and damaged mitochondria. In contrast, As-induced ROS generation promotes lipid peroxidation and protein damage, causing the loss of mitochondrial membrane potential and the consequent increase in mitochondrial membrane permeability, which in turn induces PTEN-induced putative kinase 1/parkin-mediated mitophagy [124]. Furthermore, both As^III^ and DMA^V^ promote apoptosis in cerebellar neurons, with the inorganic form being more toxic, and As^III^ -induced apoptosis may involve the activation of c-Jun N-terminal kinase (JNK) 3 and p38 mitogen-activated protein kinase (p38 MAPK) pathways [125]. Exposure to As^III^ in mice appears to be associated with increased levels of biomarkers of DNA damage and apoptosis, such as cleaved poly (ADP-ribose) polymerase-1 and cleaved caspase-3, indicating the activation of apoptosis [126]. As can also be responsible for autophagic/apoptotic neuronal cell death through the dephosphorylation of protein kinase B (Akt), which is involved in the regulation of multiple cellular functions, including cell differentiation, proliferation, survival, and apoptosis, and the phosphorylation of adenosine monophosphate-activated protein kinase, which plays a crucial role in energy and functional maintenance, survival, and apoptosis [127].As can interfere with mitochondrial oxidative phosphorylation and ATP synthesis, which, when determining a reduction in ATP levels, are responsible for ER stress and the impairment of the buffering capacity of ER Ca^2+^ channels. In addition to As-mediated oxidative stress, increased Ca^2+^ levels in intracellular compartments, including mitochondria, and the subsequent destruction of Ca signaling, lead to the activation of calpain 1, which, by inducing tau kinase activity [105,121], promotes tau hyperphosphorylation and the aggregation of tau, which is responsible for the destruction of the cytoskeletal structure of neurons.Chronic exposure to inorganic As is accompanied by an increase in the enzymatic activity of beta-site APP cleaving enzyme 1 (BACE1), resulting in the increased production of Aβ, and by the overexpression of the receptor for advanced glycation end products, leading to a disturbance in the equilibrium of Aβ clearance and, ultimately, an increased Aβ accumulation in brain tissue [128].As exposure is also associated with increased expression of pro-inflammatory markers such as IL-1β, IL-6, tumor necrosis factor alpha (TNF-α), interferon gamma, and transforming growth factor beta. In addition, As causes abnormalities of CD4+ T cell subpopulations, including polarization to T helper (Th)1 subpopulations, a decrease in Th17 cells, as well as increments of regulatory T cell populations [129,130].As may significantly reduce the activity of AChE in a dose-dependent manner, thus indicating AChE as the ideal candidate biomarker to evaluate the neurotoxic effects of As [96]. Furthermore, As can influence the concentrations of other neurotransmitters, i.e., it can increase levels of dopamine, serotonin, and their metabolites and reduce norepinephrine levels in specific brain regions [131,132].

### 2.3. Cadmium

Although rare, Cd is naturally found in the Earth’s crust, and it is generally detected in combination with ores of Pb, Cu, and Zn [133,134]. The widespread distribution of Cd in the environment is principally attributable to anthropogenic activities such as mining and smelting, the usage of high-phosphate fertilizers, municipal and sewage sludge incineration, and the industrial production of batteries, plastics, electroplating, and pigments [133,134]. Geogenic processes like weathering and erosion of rocks, volcanic eruption, forest fires, and marine aerosols may further increase Cd levels in soil and groundwater [133,135]. In addition to occupational exposure in certain industrial settings, the main sources of Cd include cigarette smoking, due to the known elevated Cd accumulation in tobacco leaves, and foodstuffs for the non-smoking general population [133,134]. For the European population, Cd dietary exposure has been estimated at 2.04 µg/kg bw per week, with grains and grain products, vegetables, and vegetable products representing the greatest contributors among food groups [136]. Looking at the individual dietary intake, it ranges from 1.15 to 7.84 µg/kg bw per week depending on dietary habits [136]. In 2011, the EFSA CONTAM Panel set a TWI of 2.5 µg/kg bw of Cd, which should ensure a high level of protection of all population subgroups, including those potentially most exposed (children, vegetarians, smokers, and subjects residing in highly contaminated areas) [137]. Following inhalation or ingestion of contaminated foods, Cd is absorbed in the lungs (from 10 to 50% of the inhaled dose) and in the gastrointestinal tract (approximately 5–10% of the ingested dose, but up to 40% in young women with low Fe stores), respectively, and enters the systemic circulation where, bound to metallothioneins, a family of small, cysteine-rich metal-binding proteins, it is distributed to various organs and tissues such as the kidneys and bones [40,138,139,140,141]. Notably, conversely to other toxic metals, Cd may cross the BBB only in small amounts, but can reach the CNS via the olfactory system [139], fostering the so-called olfactory vector hypothesis (OVH) for neurodegeneration [142]. The kidneys are the major organs in which Cd accumulates (Cd’s half-life is between 6 and 38 years); therefore, the concentration of Cd in the urine is suggestive of chronic exposure [133,134]. Beyond being classified as a group I human carcinogen [143], chronic Cd exposure correlates with increased occurrences of diabetes, osteoporosis, hypertension, decreased lung function, and kidney damage [5]. Massive concentrations of Cd in food and drinking water in Japan over the past century caused chronic Cd poisoning, known as “itai-itai” syndrome, characterized by renal tubular dysfunction and osteomalacia [49].

#### 2.3.1. Cadmium and Dementia: The Epidemiological Evidence

To date, only a small number of epidemiological studies linking Cd exposure to dementia have been performed, primarily due to the paucity of relevant exposure data [139]. In the aforementioned case–control study by Adani et al. [35], the authors aimed to evaluate the role of environmental exposure in EOD etiology, and failed to detect any association between occupational exposure to Cd and EOAD, while no subjects among the cases with EOFTD had had a Cd exposure in the workplace. In a case–control study conducted in South Korea on 89 patients with AD and 118 non-demented subjects, serum Cd levels were significantly higher in the cases than in the healthy controls; however, this association became non-significant after adjustment by age [114]. Additionally, serum Cd levels were not significantly correlated with MMSE scores, although the small number of subjects involved represents a major limitation of this study [114]. Subsequently, based on data of the 1999–2004 Third National Health and Nutrition Examination Survey (NHANES) and the Linked Mortality File (4,064 U.S. participants aged ≥60 years old and 51 of whom died of AD), a significantly positive association was observed between blood Cd levels and the risk of AD mortality (HR = 3.83, 95% CI: 1.39–10.59) after adjustment for rice eating, body mass index, diabetes, and hypertension [144]. Furthermore, in the Kaplan–Meier survival curve analysis of cumulative AD mortality by blood Cd quartile at baseline, the highest quartile (Cd concentration 0.6 μg/L) was associated with a borderline significant decrease in survival when compared with the lowest blood Cd concentration quartiles (*p* = 0.0684), suggesting the potential role of Cd in AD pathogenesis and as a predictive marker of a poor prognosis of AD [144]. Nonetheless, it should be noted that, unlike the level of Cd in the urine, which is indicative of the real body burden, blood Cd has a half-life of 3-4 months, and thus reflects recent exposure [145]. Thus, the exposure assessment in this study might be incomplete and less reliable for a chronic disease such as AD, while a 100-month follow-up period is generally too short to evaluate AD survival related to different exposures [144]. Another population-based study attempted to address this relevant issue by separately testing the association between urinary Cd measured at NHANES baseline interviews (cycle 1999–2000 through cycle 2005–2006) and NHANES III (interviews in 1988–1994) and AD mortality over the next 5 to 13 years [146]. In the first analysis, an interquartile range (IQR) = 0.51 ng/mL increase in urinary Cd was associated with a 58% increase in AD mortality (HR = 1.58, 95% CI: 1.20–2.09, *p* = 0.0009; mean follow-up = 7.5 years) after adjustment for confounding factors (ethnicity, sex, smoking status, education, and urinary creatinine) and, although weaker than urinary Cd, an IQR increase in blood Cd was also significantly associated with increased AD mortality (HR = 1.22, 95% CI: 1.01–1.48, *p* = 0.04; mean follow-up = 13 years) [146]. In the second analysis conducted on NHANES III participants, who were comparable to those from NHANES 1999–2006 in terms of baseline covariates, urinary Cd was related to an increase in AD mortality (HR = 1.11, 95% CI: 1.02–1.20, *p* = 0.0086; mean follow-up = 12.7 years) without adjustment for urinary creatinine; however, both in all models considering a 23-year follow-up and in a shortened follow-up period after adjustment for creatinine, the significant positive association was no longer observable [146]. Therefore, if these findings confirm the positive association between blood Cd and AD mortality shown by [144], and support the role of urinary creatinine as an independent risk factor of AD mortality, they also suggest greater caution in the assertion of a Cd–mortality association in general, and also in the prospect of a lack of consensus on the possibility that low levels of urinary Cd can vary widely with recent exposure and are associated with age through a non-linear and non-monotonic relationship [146,147]. In addition, the authors raise doubts about the utility of urinary creatinine adjustment and recommend the need to confirm these results through toxicological studies capable of providing insights into the Cd–mortality relationship. Finally, a meta-analysis of eight studies including 405 AD patients and 424 control subjects reported that subjects with AD had significantly higher circulatory Cd concentration than controls (SMD = 0.62, 95% CI: 0.12–1.11, *p* = 0.0144), with a significant heterogeneity between studies but no publication bias detected [73]. Notably, since the population studies were from Asia and Europe, the authors did not exclude that Cd exposure in these populations could be attributable to smoking [73]. More recently, despite measuring a significantly increased concentration of Cd in 50 patients with AD than in age-matched controls (*p* < 0.0001), Yadav and co-workers [74] observed no correlation between Cd levels and the expression of selected genes that are upregulated in patients with AD, probably as a consequence of the small sample size. As already shown for Al, the case–control study by Babić Leko et al. [48], conducted on a total of 193 Croatian participants, reported a significantly positive correlation between Cd levels in CSF and levels of CSF biomarkers of AD, namely phosphorylated tau isoforms, VILIP-1, PAPP-A, and albumin.

In summary, a relatively large body of epidemiological evidence suggests the potential involvement of Cd in AD etiopathogenesis, with elevated body Cd levels apparently predictive of an increased mortality risk for AD. However, some criticisms still need to be solved (Table 4). Therefore, further longitudinal population-based studies are warranted to definitely establish the most relevant window of exposure and consequently to accurately estimate the period of latency between Cd exposure and the development of AD and/or mortality caused by AD. Future studies should use urine specimens as the gold standard to determine the actual body Cd load, possibly with repeated measurements over time, although the issue regarding urinary creatinine adjustment requires more in-depth research. An attempt to assess multiple environmental exposures could provide the potential contribution of other risk factors to AD’s etiopathogenesis. Finally, the association between Cd exposure and the risk of FTD is currently and surprisingly almost unexplored, and, given the individual and social impact of this disease and the various manners of Cd exposure, studies investigating this relationship would be desirable.

#### 2.3.2. Cadmium Neurotoxicity: The Underlying Mechanisms

Similarly to the other toxic metals examined, several major pathways through which Cd can increase the risk for neurodegenerative disease have also been described (reviewed in [148,149]):Although Cd does not directly generate ROS via redox reactions, a growing body of evidence suggests that it can enhance the production of superoxide radicals at the mitochondrial level through a direct role in Complexes I, II, and III of the electron transport chain (ETC), inhibiting electron flow rather than uncoupling oxidative phosphorylation, as has long been believed to occur. Cd may also induce ROS generation by opening the permeability transition pore (PTP), a protein complex located at the interface between the inner and outer membranes in mitochondria. The opening of the PTP leads to the alteration of the mitochondrial membrane potential and the consequent impairment of ATP synthesis, as well as to the release of cytochrome c reserves [148,149]. Cytochrome c, being responsible for the electron transfer from Complex III to Complex IV, impairs the ETC and further promotes the generation of ROS, as well as representing a signal for apoptotic cell death [148,149]. Notably, the exposure of neuroblastoma cells to Cd results in a dose- and time-dependent increase in intracellular ROS levels [150]. Due to its affinity for thiol groups, Cd may also induce GSH depletion, causing intracellular oxidative stress, and the disruption of thiol homeostasis, which can ultimately lead to cell death, as observed in primary cultures of oligodendrocytes and cortical neurons [151,152]. Cd has also been recently shown to induce oxidative stress and mitochondrial dysfunction by inhibiting the expression of sirtuin 1 (SIRT1) in PC12 cells and primary rat cerebral cortical neurons [153]. SIRT1 is a potent intracellular suppressor of oxidative stress, which stimulates the expression of antioxidants (SOD, CAT, thioredoxin) and regulates immune responses via NF-κB signaling [154]. Notably, SIRT1 has been demonstrated to play a crucial role in neurogenesis, synaptic plasticity, memory development, and the protection of neurons from death and degeneration [155].Cd may impair autophagy by inducing the expansion of autophagosomes with a concurrent upregulation of autophagy-related proteins (i.e., LC3-II and p62) in PC12 cells and primary murine neurons. In particular, Cd induces Ca^2+^-dependent activation of JNK and Akt pathways, two major regulators of neuronal survival, as well as of Beclin-1, an essential autophagy protein and a target of Akt. Indeed, Akt-mediated phosphorylation of Beclin-1, triggering the dissociation of Beclin-1 from Bcl-2, can activate autophagy via Ca^2+^/calcium/calmodulin–serine/threonine kinase death-associated protein kinase [156,157,158,159].Cd influences neurotransmission through the desynchronization of the neurotransmitter release, which is accompanied by a rapid rise in mitochondrial ROS production and lipid peroxidation, and by a decrease in the concentration of neurotransmitters available for each event, with the latter especially observed in glutamatergic neurons. Furthermore, Cd can downregulate the expression of M1, M2, and M4 cholinergic muscarinic receptor genes in the frontal cortex and hippocampus [160]. It is of interest that the selective blockade of muscarinic M1 receptors in cholinergic neurons from the basal forebrain, associated with impaired memory and learning deficits, is mediated by oxidative stress, overexpression of glycogen synthase kinase 3 beta (GSK-3β), impaired expression of AChE splice variants, and increases in the production of Aβ protein and phosphorylated and total tau protein, and ultimately leads to cell death [161,162].Cd appears to impair the functionality of glycogen phosphorylase (GP) enzyme, a key enzyme of glycogenolysis, the process by which cellular stores of glycogen are catabolized into glucose, resulting in the accumulation of glycogen in CNS cells, primarily in glial astrocytes, which represents a relevant mechanism of neurological symptoms. GP is rich in cysteine residues, which are sensitive to binding metals such as Cd. In addition, GSK-3β is considered a tau kinase, and its phosphorylation promotes hyperphosphorylation of tau protein [149,163].Finally, Cd affects Zn homeostasis through the downregulation of the Zn transporter ZnT3, which is primarily localized at glutamatergic synapses in the hippocampus and neocortex, where it fosters Zn’s release into the synaptic cleft [149,164]. Given the beneficial effects of Zn on learning and memory and in modulating synaptic transmission and plasticity, ZnT3 may regulate cognitive function [165]. Additionally, Zn deficiency may have implications for AD, as observed in human postmortem brain tissue from individuals with AD, which were characterized by a significant loss of ZnT3 expression in cortical regions compared to control subjects, as well as in transgenic mice, in which a low-Zn diet was associated with a substantial increase in plaque volume [164,166].

### 2.4. Lead

Among heavy metals, Pb is considered the second most toxic element; it is naturally found in very limited amounts in the Earth’s crust (0.002%) and is mostly released into the environment by human activities, including the use of fossil fuels, some types of industrial plants and contaminated sites (e.g., Pb smelters), the past use of Pb-based paints, and the applications of a wide range of products like batteries, cosmetics, ceramic glazes, jewelry, toys, ammunition, and solders [167,168,169]. Numerous uses of Pb in the last century, due to its malleability and corrosion resistance, have allowed the release of this dangerous metal in all environmental matrices, with a consequent high risk of exposure for humans [169,170]. Pb exists in its inorganic form, mainly contained in paint, soil, dust, and various consumer products, and in organic forms—tetraethyl and tetramethyl Pb—used in leaded gasoline, which, however, was banned in Europe in the 1990s [171,172]. Although their exposure is currently limited to an occupational context, organic forms of Pb are extremely harmful since, once ingested or absorbed through the skin, they exert their toxic effects on the CNS [172]. As regards inorganic forms, although the causes and extent of exposure depend on the geographical areas, the inhalation of Pb dust in working environments and of Pb-based paint undergoing deterioration or during the renovation of older buildings and its ingestion via water (from leaded pipes) and food (from lead-glazed containers) and due to hand-to-mouth behavior in children are the most common routes of exposure [78,168,169]. Pb absorption in the organism depends on a variety of factors, including the chemical form of the Pb, the route of exposure, and the nutritional status, health, and age of the individual [169]. While most inorganic inhaled Pb is absorbed in the lower respiratory tract, adults normally absorb from 20% up to 60–80% of ingested inorganic Pb, depending on fasting conditions, and it is excreted in the urine or through biliary clearance [172]. Unlike blood, where Pb has a half-life of 28 days, up to 95% of Pb resides in the skeletal system (with a half-life of decades), from which the metal is slowly released into the circulation [172,173,174]. In 2021, The World Health Organization (WHO) estimated that approximately half of the 2 million deaths attributable to chemical exposures in 2019 were associated with Pb exposure [175]. Additionally, Pb exposure is estimated to account for 30% of the global burden of idiopathic intellectual disability [168]. Although a safe level has not been identified, yet, blood Pb concentrations ranging from 5 µg/dL to 10 µg/dL have been associated with a variety of effects, including neurocognitive and behavioral developmental disorders in children and cardiovascular disease in adulthood; therefore, the WHO set 5 µg/dL as the threshold value at which to start clinical intervention [176]. In addition to neurological and cardiovascular effects, Pb exposure may cause renal, reproductive, and endocrine toxicity, and, due to its ability to cross the placenta, maternal Pb can affect fetal development [172]. Cereals, vegetables, and tap water are the major sources of Pb exposure in European adults, for whom the mean Pb intake has been estimated to range between 0.36 and 1.24 µg/kg bw per day; elderly and very elderly population groups show similar profiles, while high consumers have an intake of up to 2.43 µg/kg bw per day [177,178]. In 2010, the EFSA CONTAM Panel concluded that the provisional TWI of 25 μg/kg bw was no longer precautionary due to the lack of evidence for a threshold for Pb toxicity [178].

#### 2.4.1. Lead and Dementia: The Epidemiological Evidence

As already highlighted for other metals, there are little data on the association between Pb exposure and the risk of FTD. The case–control study by Adani et al. [35] documented a positive and an inverse correlation, albeit non-significant, between occupational Pb exposure and EOD (OR = 1.09, 95% CI: 0.23–5.18) and EOFTD (OR = 0.41, 95% CI: 0.04–4.02), respectively. Current epidemiological research on the role of Pb in AD is also limited. A systematic review of six case–control studies showed no evidence to support any association with occupational Pb exposure, despite the overall low quality of the investigations [179]. A more recent systematic review, including 12 case–control studies measuring Pb concentrations in various biological specimens, reported conflicting results, with no significant correlations with CSF, blood, and nail Pb levels and an inverse association with CSF and hair Pb [70]. In the 2000s, a series of studies were conducted on subgroups in the Normative Aging Study (NAS), a multidisciplinary longitudinal study of aging in elderly men, in order to investigate whether Pb can modify MMSE scores. The first of them, based on a cross-sectional design and including 733 participants, showed that with an increase from the lowest to the highest quartile of blood Pb, tibia Pb, and patella Pb, the ORs were 3.4 (95% CI: 1.6–7.2), 2.2 (95% CI: 1.1–3.8), and 2.1 (95% CI: 1.1–4.1), respectively, for an MMSE score < 24 (a traditional cut-off point for dementia) [180]. Furthermore, for each year of increase in age, the highest quartiles of patella or blood Pb were associated with a 4-fold decrease in MMSE score if compared to the effect of age in the lowest quartile of Pb levels, indicating that Pb exposure can modify the effect of age in cognitive decline [180]. A subsequent longitudinal study within the NAS that included 466 male elderly subjects found that a one-interquartile range (corresponding to 20 µg Pb/g of bone mineral) higher patella Pb concentration was associated with a decreased MMSE score (−0.24, 95% CI: −0.44–−0.05) after adjustment for confounders (age, educational level, smoking, alcohol intake, and time between the two MMSE tests) [181]. A weaker association was also measured with tibia Pb, but no association of MMSE score change was observed with blood Pb concentration, with the overall results confirming that cumulative Pb exposure, even at relatively low levels, may predict a decline in cognitive function among non-occupationally exposed elderly men [181]. In contrast to the study by Wright et al. [180], however, the authors did not report any interaction between Pb and age in MMSE score decreases over an average period of 3.5 years [181]. In a more recent characterization of the association between bone Pb concentration and variation in cognition over time in a subcohort of the NAS (741 subjects with MMSE scores, 715 subjects with global cognition summary scores) undergoing up to five visits over 15 years, bone Pb levels, a proxy of chronic exposure to Pb, were confirmed to be linked to a faster decline in MMSE scores [72]. Indeed, beyond being significantly associated with lower baseline MMSE scores (−0.13, 95% CI: −0.251–−0.004; *p* = 0.04), an IQR (corresponding to 21 µg Pb/g of bone)-higher level of patella Pb was associated, on average, with a lower MMSE score by 0.016 points per year (95% CI: −0.032–−0.0004; *p* = 0.04), and similar but weaker associations were also found for tibia Pb [72]. Importantly, an IQR increase in patella Pb was associated with an increased risk of having an MMSE score < f 25, although with a borderline statistical significance [72]. On the other hand, the association between patella Pb and global cognition score (the sum of scores obtained from seven individual tests in the Neurobehavioral Evaluation System 2, Consortium to Establish a Registry for Alzheimer’s Disease and the Wechsler Adult Intelligence Scale—Revised), at baseline and over time, was not significant, and a similar, albeit weaker, result was found for tibia Pb [72]. Notably, the half-life of Pb in the skeleton varies by site and, while tibia Pb represents a measure of cumulative metal exposure, patella Pb may be considered as a source of body Pb burden that can be mobilized into the circulation, thus explaining the stronger association observed with MMSE score for patella Pb compared to tibia Pb [72]. The systematic review and meta-analysis by Xu et al. [73], including ten studies for a total of 716 AD patients and 1298 control subjects, revealed that AD subjects had lower circulating Pb levels than healthy controls (SMD = −0.23, 95% CI: −0.38–−0.07; *p* = 0.0043); however, a significant heterogeneity was observed between studies. A more recent systematic review and meta-analysis, aimed at evaluating the exposure to Pb in subjects with AD and controls based on measurements in various biospecimens, identified 15 case–control studies that used blood (seven used whole blood, eight used serum; five used CSF, three used hair/nails, three used postmortem brains, one used urine and none used bone [182]). The two distinct meta-analyses separately performed on studies assessing Pb in whole blood and in serum did not find significant differences in Pb levels between groups, contrary to the findings of Xu et al. [73,182]. The included studies assessing Pb in other biological matrices reported conflicting results, with no or an inverse association of Pb levels with AD [182]. Notably, although blood and serum Pb are generally used as biomarkers of recent exposures, we cannot exclude that blood Pb also reflects past exposure to Pb and other individual factors, giving rise to uncertain and mixed results regarding Pb’s association with AD, while serum Pb, given its extremely low concentration, can be more subject to contamination errors [182]. As occurs with blood, hair, and nail measurements are indicative of current or recent Pb exposure, while the lack of a significant association between Pb levels in postmortem brains and AD may suggest that Pb could play an etiological role in AD, with no long-term deposition in the brain [182]. An Indian case–control study performed on 50 subjects with AD and 50 age-matched non-demented control subjects confirmed the previous findings and observed no differences in whole blood Pb concentrations between the two groups, with AD patients free of trace amounts of circulating Pb [74]. Finally, a case–control study aimed at evaluating the involvement of metals in AD pathogenesis that was conducted on healthy, mildly cognitively impaired, and AD subjects revealed a significant positive correlation between CSF Pb and the expression levels of phosphorylated tau, VILIP-1, PAPP-A, and albumin, which are characteristic biomarkers of AD [48].

Overall, while Pb is widely known to exert a variety of neurotoxic effects (see the following section), the literature exploring the role of Pb in the onset of dementia is scattered. Some signals of a potential contribution of Pb to the etiopathogenesis of AD have been reported by the few studies evaluating the correlation between Pb levels in bone (patella or tibia) and cognitive function, both at baseline and over time, in cohorts of elderly subjects with AD. On the other hand, most of the published studies to date have primarily used blood or serum Pb levels, which reflect recent exposure and, therefore, cannot be linked to early AD-related pathological changes. Consistently, no significant differences were observed in circulating Pb levels between AD subjects and healthy controls, suggesting the need to use bone as the gold-standard biological sample within case–control studies to definitely ascertain the involvement of this metal in the onset of AD and, in general, dementia (Table 5).

#### 2.4.2. Lead Neurotoxicity: The Underlying Mechanisms

Pb has been widely recognized as a neurotoxicant able to disrupt the CNS through multiple cellular and molecular mechanisms (reviewed in [163]):Pb can exert ionic toxic effects due to its ability to replace other divalent cations like Ca^2+^, magnesium, and iron and monovalent cations like sodium (Na^+^). In particular, by replacing Ca^2+^, Pb may cross the BBB and, through VGCCs, it reaches astroglia and neurons, causing severe damage to the prefrontal cerebral cortex, hippocampus, and cerebellum. Pb may also affect Na+ concentrations, thus leading to the dysfunction of Na^+^-dependent processes, including the generation of action potentials in the excitatory tissues, uptake of neurotransmitters (choline, dopamine, and gamma-aminobutyric acid), and modulation of uptake and storage of Ca^2+^ by synaptosomes [183,184].Pb interferes with the transmission of glutamate, the most common neurotransmitter in the brain, by acting as a non-competitive, voltage-dependent antagonist of the NMDA receptor, a non-selective ion channel for cations, whose action is related to synaptic plasticity and to the phenomenon of long-term potentiation (LTP), a process associated with the acquisition of information. In Pb-exposed rats, synaptic transmission of both NMDA and α-amino-3-hydroxy-5-methyl-4-isoxazolepropionic acid (another ionotropic receptor for glutamate, permeable to both Na^+^ and Ca^2+^) was significantly reduced, which in turn caused LTP impairment and morphological changes in the dendrites of CA1 pyramidal neurons, ultimately resulting in deficits in learning and memory [183,185].Although, like Cd, Pb is a redox-inactive metal, it can induce oxidative stress through increased ROS generation and GSH depletion by binding to sulfhydryl groups. In addition, Pb inactivates antioxidant enzymes such as SOD, CAT, GPx, GR, and glutathione-S-transferase, causing a further reduction in GSH reserves [184,186]. Like other toxic metals, Pb accumulates in the mitochondria and, by disrupting Ca^2+^ homeostasis, may favor the increase in intracellular Ca^2+^, which, by enhancing mitochondrial ETC, can lead to an increase in ROS production [183]. A recent study also revealed that neuronal cells exposed to Pb and Aβ were characterized by a depolarization of the mitochondrial membrane potential, a decrease in ATP levels and mtDNA copy numbers, and the disruption of ETC through a reduced expression of genes and proteins of the ETC complex [187].Both acute and chronic exposure to Pb may enhance the expression of AD-related genes and their products—APP and BACE1—with the latter being an aspartyl protease of the pepsin family performing the β-cleavage of APP that, in turn, generates amyloidogenic peptides and possibly Aβ plaque deposition in brain tissues [188,189]. Furthermore, Pb exposure has been associated with a reduced expression of low-density lipoprotein receptor-related protein-1, a member of the LDL receptor family located in the BBB, which is involved in the Aβ transport from the brain to the blood, thus promoting a further Aβ accumulation in the brain [190].Perinatal and neonatal Pb exposure, even at concentrations considered “safe for humans”, upregulates tau protein and promotes tau hyperphosphorylation at several serine and threonine residues by stimulating the activities of the two major tau kinases, i.e., GSK-3β (through phosphorylation) and cyclin-dependent kinase (through calpain-catalyzed proteolysis), processes collectively responsible for tau aggregation and the formation of NFTs, cytoskeleton disruption, and subsequent cell death [191,192].Neurotoxicity of the CNS can be attributed primarily to the neuroinflammatory response and neuronal cell death in the hippocampus and forebrain. Indeed, by inducing mitochondrial Ca^2+^ overload and the consequent generation of mitochondrial ROS, Pb can activate the NLR family pyrin domain containing 3 (NLRP3) inflammasome in microglia, which in turn leads to the cleavage of caspase-1 and the release of IL-1β and IL-18 and, ultimately, to the inflammatory cascade. Notably, microglia and NLRP3 inflammasome activation are related to learning and memory impairment in mice [193,194]. Pb exposure can also trigger the toll-like receptor 4-MyD88-NFκB signaling pathway, which, by mediating microgliosis, astrogliosis, and the abnormality of neuronal genesis, leads, as downstream effects, to increased expression levels of IL-1β, TNFα, p38 MAPK, and ERK1/2 [195].Pb is also recognized as an epigenetic modifier that exerts effects at diverse levels, including DNA methylation and modifications of histones (methylation and acetylation) and non-coding RNAs (ncRNA: microRNAs, long non-coding RNAs, circular RNAs), which are known regulators of processes such as oxidative stress, inflammation, and neuronal apoptosis that are involved in the development of neurodegenerative diseases (reviewed in [196,197]). In particular, DNA methylation, which occurs under the action of DNA methyltransferases (DNMTs), is the most studied and stable epigenetic mechanism and may even persist into the next generation. Long-term Pb exposure was associated with reduced activity levels of DNMT1, DNMT3a, and methyl-cytosine-phosphate-guanine (Me-CpG)-binding protein-2 (an epigenetic regulator binding methylated cytosines at the CpG site on DNA) [196]. In addition to overall altered DNA methylation levels and DNMT1 expression, Pb exposure appears to induce hypermethylation of the APP promoter gene, indicating that the metal can be implicated in the pathogenesis of AD [196]. Regarding the mechanisms of Pb toxicity in AD based on histone modifications and alteration of ncRNA expression, the evidence is rather scarce and requires further exploration in future investigations [196].

### 2.5. Mercury

Hg is a toxic element naturally present in the Earth’s crust and, although events such as volcanic activity and weathering of rocks contribute to its release into the environment, human actions, including coal-fired power plants, residential coal burning, industrial processes, waste incinerators, and mining, represent the main sources of Hg [198]. Hg exists in three different forms: elemental or metallic Hg, used in older thermometers, which, at room temperature, can be released easily into the atmosphere as an invisible, odorless toxic vapor; inorganic Hg salts, generally in the solid state as mercurous or mercuric salts and in Hg compounds with chlorine, sulfur, or oxygen, which can be subsequently weathered from rocks and, transported in water, also found in soil; and methylmercury (MeHg), produced from the methylation of inorganic Hg by microorganisms present in the environment, which is the most common toxic Hg compound [199,200]. In addition to its bioaccumulation in fish and shellfish, MeHg also tends to biomagnify in aquatic ecosystems; thus, its concentration is much higher in large predatory fishes than MeHg absorbed in smaller fishes, which in turn have acquired Hg through the ingestion of plankton [198]. Routes of exposure and effects in humans differ between Hg forms. Elemental Hg enters the body mainly through inhalation during industrial processes and, after absorption in the lungs (absorption rate about 80%), is oxidized into mercuric ions in the blood and distributed to various organs [198,199]. Being lipid soluble, metallic Hg may cross the blood–placenta barrier and BBB, although it may also reach the brain and accumulate in the CNS via the olfactory pathway [142,199]. Conversely, the primary manner of exposure to inorganic Hg is represented by ingestion, with an absorption rate between 2 and 38% in the gastrointestinal tract, and subsequent accumulation of Hg salts in the kidney [199,201]. However, exposure in the general population occurs mainly through the consumption of fish meat (tuna, swordfish, cod, whiting, and pike) and fish products contaminated with MeHg, which is absorbed almost completely in the digestive tract (with a rate 17 to 35 times faster than that of inorganic Hg) and, like elemental Hg, easily crosses the BBB and blood–placenta barrier and accumulates in the brains of fetuses more than in those of mothers [201,202]. The biological half-life of inorganic Hg is around 60 days, while elemental Hg, binding to selenium or sulfhydryl groups, can be detected in the brain for a long time, with an estimated half-life of up to 20 years [199]. Both Hg forms are excreted in the urine, and elemental Hg is also excreted in feces, breath, sweat, and saliva [199]. MeHg has a half-life of 70 days and is mostly excreted in the bile and, after demethylation, in the feces [201]. The EFSA CONTAM Panel set a TWI of 4 µg/kg bw and 1.3 µg/kg for inorganic Hg and MeHg, respectively [202]. As for MeHg, the average dietary exposure across age groups does not exceed the TWI, except for toddlers and high-fish consumers, who may exceed the TWI by up to approximately six times [202]. Both elemental Hg and MeHg produce toxic effects on the central and peripheral nervous systems, and the inhalation of Hg vapor also generates toxicity to the digestive and immune systems, lungs, and kidneys [198]. It is important to remember that the high consumption of MeHg-contaminated fish was responsible for the Minamata disease that occurred in Japan in the 1950s, which manifested in auditory and visual disorders, ataxia, dysarthria, and tremor [49,203]. Inorganic Hg salts are instead harmful to the skin, eyes, and gastrointestinal tract and, in the case of ingestion, to the kidneys [198].

#### 2.5.1. Mercury and Dementia: The Epidemiological Evidence

So far, the association between Hg exposure and the onset of FTD has not been explored, while a limited number of studies have focused on the effects of Hg on cognitive function and AD development. A systematic review including 13 studies (12 case–control and 1 longitudinal controlled cohort) reported a dose–response relationship between the urinary Hg concentration and a decrease in cognitive abilities (memory and attention measures) in workers currently exposed to high doses of Hg in industry or mining, although some heterogeneity was observed across studies regarding clear cut-off points in Hg excretion between cases and controls [204]. The analysis of nine studies (five retrospective cohorts, four case histories) that assessed past occupational exposure (dating back from 5 to 30 years) to high-dose Hg showed, overall, little evidence of a negative association between Hg exposure (generally determined by urine Hg concentration) and current scores on neurological tests and clinical symptoms [204]. The authors also evaluated the relationship between Hg exposure and AD from a total of 16 studies (7 on living patients, 9 based on postmortem autopsy): the studies on living subjects, which quantitatively measured Hg in various biospecimens, i.e., urine, plasma, hair, nails, and autopsy studies, documented conflicting results, with a positive association (increased Hg levels in blood, urine, and various areas of the brain in patients with AD), an inverse correlation (lower Hg concentrations in the blood and brain in AD subjects compared to controls), or a lack of association (in the brain), mainly due to the small sample size and evaluation of different brain areas [204]. Interestingly, one prospective cohort included in this systematic review found that nail Hg concentrations decreased with increasing age and severity of AD in patients, likely because of a progressively reduced ability to excrete Hg in subjects with more severe dementia [205]. Therefore, while findings from studies in workplace settings suggest an effect of Hg on clinical signs relevant to AD, it is unclear whether cognitive deficits may lead to dementia later. The systematic review by Cicero et al. [70] reported that 6 out of 11 case–control studies found significantly higher levels of Hg in AD patients than in healthy subjects and, as for the other biospecimens, CSF Hg concentrations were not significantly different between AD and control subjects, nail Hg levels were significantly decreased in patients with AD in the two included studies, and the studies measuring hair Hg observed inconsistent results. A systematic review and meta-analysis of seven studies (overall, 397 control subjects and 478 AD patients) aimed at evaluating Hg levels in the circulation (blood, serum/plasma) estimated a significantly higher Hg concentration in AD patients than controls (SMD = 0.55, 95% CI: 0.15–0.95, *p* = 0.0073), although the small number and heterogeneity of investigations included are among the major limitations of this study [73]. The aforementioned study by Adani et al. [35] reported a non-significantly decreased risk of EOAD in relation to occupational exposure to Hg (OR = 0.80, 95% CI: 0.06–10.49); however, as previously noted, no conclusions can be drawn due to the limited number of subjects involved. More recently, Yadav and co-authors [74] documented significantly higher concentrations of Hg in the whole blood of 50 AD patients compared to a group of age-matched non-demented subjects (*p* < 0.0001); however, as for Al and Cd, they failed to find a correlation between the metal level and the expression of selected genes upregulated in AD, namely APP, presenilin 1 (PSEN1), and presenilin 2 (PSEN2), with PSEN1,2 being the major components of ϒ-secretase that mediate the final cleavage that releases Aβ [206] and APOE4, which contribute to the generation of Aβ oligomers in the brain [207]. Conversely, Babić Leko et al. [48] demonstrated that plasma Hg concentration was positively correlated with the levels of other biomarkers of AD measured in CSF.

In summary, despite the general paucity of data from longitudinal studies, a sufficient amount of evidence supports the deleterious effects of continuous exposure to Hg in working environments on cognitive domains, such as memory and attention, that may also be related to early clinical signs of AD; however, it remains to be clarified whether occupational exposure to high or even low doses of Hg is also associated with the risk of AD. In contrast, there are only some signals of association between Hg exposure, measured as blood Hg concentration, and risk of AD in the general population (Table 6). Therefore, future long-term studies on large cohorts of patients from all world regions should be conducted to observe the potential transition from cognitive impairment to full-blown dementia associated with Hg exposure from both environmental and occupational sources. Importantly, the relationship between Hg exposure and FTD is currently totally unstudied; this gap should be addressed soon with high-quality research.

#### 2.5.2. Mercury Neurotoxicity: The Underlying Mechanisms

Hg neurotoxicity has been long known, and a plethora of mechanisms, in particular with reference to MeHg, have been proposed:MeHg is considered a potent pro-oxidant capable of interfering with both enzymatic and non-enzymatic antioxidant molecules. It can directly inhibit Cu-Zn SOD and CAT and promote a decrease in neuronal GSH levels by impairing the transport of GSH precursors and subsequent GSH synthesis in astrocytes. Due to its ability to react with sulfhydryl groups, MeHg may selectively inactivate Mn-SOD and further reduce the availability of the GSH pool. Additionally, Hg has a great affinity for selenothiol groups of enzymes such as GPx, thioredoxin reductase, and selenoprotein W, which are involved in antioxidant defense, and can directly inhibit their activities [208,209]. MeHg may also interfere with Complexes I, II, and III of the mitochondrial ETC activity, resulting in an increased production of superoxide anions and H_2_O_2_ [210,211].MeHg can induce programmed cell death. Oxidative stress generated in mitochondria causes a decrease in membrane potential and increased membrane permeability, which induces the release of cytochrome c, an activator of caspase-mediated apoptosis. Alternatively, the covalent binding of MeHg to PTEN, a negative regulator, prevents the phosphorylation of Akt and its downstream factor CREB, thereby leading to disruption of Akt/CREB/Bcl-2 signaling and the subsequent induction of proapoptotic signaling [208,212].MeHg may also promote mitochondrial dysfunction by generating excessive mitochondrial Ca^2+^ influx through the activation of Ca^2+^ channels, thus causing an impairment of ATP synthesis [213]. Importantly, together with oxidative stress, the intracellular Ca^2+^ imbalance is, in turn, responsible for tau hyperphosphorylation [214]. Furthermore, exposure to MeHg in mice was shown to induce hyperphosphorylation of tau at sites in the cerebral cortex consistent with the phosphorylation patterns observed in FTD and AD, while the distribution of phosphorylated tau corresponded with areas where the most substantial neuropathological changes occurred, i.e., there was a decrease in the number of neurons, an increase in the number of migratory astrocytes and microglia/macrophages, necrosis, and apoptosis [215]. MeHg can further activate the p44/p42 MAPK, p38 MAPK, and protein kinase C/CREB pathways, which are followed by upregulation of c-fos, a marker of neuronal activity and apoptotic neuronal cell death [216,217].MeHg can alter glutamatergic neurotransmission by both enhancing the release of glutamate from the presynaptic terminal and reducing its uptake by astrocytes, which allows more glutamate to activate NMDA-dependent Ca^2+^ channels. Dyshomeostasis of glutamate, an excitatory neurotransmitter, and intracellular Ca^2+^ increase the capacity of MeHg to induce tau phosphorylation, which, as reported above, is an NMDA receptor-independent mechanism [208,214].The toxic effects of MeHg can also target, although to a lesser extent than glutamate, catecholaminergic neurotransmission, the deterioration of which is implicated in various neurological and neuropsychiatric disorders, as it is related to crucial neuronal pathways, including cognition, memory, and motor function. Overall, MeHg induces concentration-dependent increases in dopamine release, inhibits the activity of monoamine oxidase, an enzyme involved in the catabolism of dopamine, and impairs dopamine’s affinity for brain D1 and D2 receptors. Although the effects of MeHg on the adrenergic system have been less investigated, MeHg has been ascertained to induce an increase in norepinephrine uptake, resulting in an increase in the neurotransmitter in some CNS areas such as the cerebellum, spinal cord, and caudate putamen [218].Hg and MeHg can increase levels of Aβ in both a dose- and time-dependent manner. Indeed, they both induce an overproduction of APP (which is accompanied by ROS production and glia activation) without activation of BACE1 and reduce the expression of neprilysin (NEP), the major Aβ-degrading enzyme in the brain [38,219]. Notably, NEP is reduced and inversely correlated with levels of CSF Aβ and tau in AD patients, and downregulation of NEP is associated with the progression of AD neuropathology, supporting the key role of this enzyme in protection from AD neuropathology [220,221].

## 3. Artificial Intelligence, Neurodegeneration and Heavy Metals: (Still) So Far, (Potentially) So Near?

The global development and adoption of Artificial Intelligence (AI) methods and principles have gained momentum in every portion of our everyday life, and they have been particularly influential when it comes to the biomedical framework, where the application of Data Mining, machine learning, Deep Learning, and, more recently, Generative AI principles has truly skyrocketed, especially in specific sectors like cancer research, biomedical imaging, and medical data processing. Moreover, the sectors of neuroscience and FTD clinical research have been largely and positively affected by the application of such methods. This is also the case with the application of these methods to investigations of the etiopathology and pathophysiology of FTD and related disorders, which is the main focus of the present investigation in clinical terms.

It is, indeed, a common belief that the discovery of new biomarkers for FTD and similar conditions, as well as the optimization of measurement technologies, can largely benefit from the use of AI tools, especially when used to properly carry out multivariate analyses related to multiple biomarkers at a time. This approach can be applied to the early-stage detection of FTD and other neurodegenerative disorders, but it is also useful for differential diagnoses and the stratification of patients, as well as for their monitoring, nowadays always more settled within residential frameworks, and for their prognostic assessment, to ultimately improve their quality of life [222]. However, the link between AI use, FTD (or other neurodegenerative conditions), and heavy metal exposure is still unexplored: in the next section, we seek to present the actual literature in terms of AI applications in FTD and clinically similar conditions, as well as in terms of the use of AI in studying toxic metal exposure’s effects on human health, creating the foundations for future applications merging this triad of critical scientific elements.

### 3.1. AI Applications in FTD: Diagnostic Purposes

Differential diagnosis is one of the specific fields where the employment of AI methodologies, mainly referring to those related to machine learning (ML) methods, is more largely observed within FTD investigations. In that, neuropsychological assessment represents one of the most popular approaches used to distinguish FTD from neurodegenerative conditions with similar clinical features, as well as between FTD subtypes. In this regard, AI approaches can also help in selecting the number of tests to be eventually administered to the patient, reducing the burden for the individuals and optimizing time slots for the clinical neuropsychologists and similar figures. According to Garcia-Gutierrez and co-authors, the Free and Cued Selective Reminding Test, verbal fluency tests, and Addenbrooke’s Cognitive Examination turned out to be highly predictive in the framework of differential diagnoses between bvFTD and Alzheimer’s disease (AD), with an accuracy of over 84%. Such results were obtained by trying different ML approaches including, among others, Naïve-Bayes and Support Vector Machines (SVMs) [223].

Similarly, the ML-based software Neurominer was employed to compare patterns of bvFTD, AD, and schizophrenia, to estimate predictability in patients with bvFTD and schizophrenia based on various kinds of data, including biological, clinical, and sociodemographic ones, as well as to examine genetic features, prognostic aspects, and progression in patients with clinically relevant features of psychosis or recent-onset depression. The study found that the bvFTD pattern was more largely expressed in individuals with schizophrenia and major depression than in patients with temporo-limbic AD patterns, with a bvFTD expression that was mainly predicted by high body mass index, the worsening of psychomotor features, affective disinhibition, and paranoid ideation. Furthermore, the schizophrenia pattern was quite prevalent among patients with bvFTD and was linked to the C9orf72 variant, oligoclonal banding in the cerebrospinal fluid, cognitive impairment, as well as younger age. Taken together, such retrievals demonstrated that neurobiological links likely occur between bvFTD and psychosis, directing the attention towards prefrontal and salience system alterations [224].

Brain magnetic resonance imaging (MRI) data were also employed to distinguish between FTD and AD through a combined unsupervised and supervised ML approach. Subcortical gray matter volumes and cortical thickness measures were extracted from MRI images, and dimensionality reduction was then applied to build up a single feature that could subsequently be used for classification according to an SVM. Cross-sectional data allowed researchers to reach a discrimination accuracy of 82.1% between FTD and controls, which increased to 88% when they used longitudinal data, whereas FTD was successfully distinguished from AD in 63.3% and 75% of cases with cross-sectional and longitudinal data, respectively [225]. Relevant results were also obtained by another study, which also used brain MRI and semantic fluency, highlighting regional voxel differences existing between bvFTD and controls by means of the Random Forest (RF) ML model. The research demonstrated a very high validation accuracy, ranging from 88 to 91%, using only MRI and adding semantic fluency, respectively, in the validation using the bvFTD cohort [226].

Differential classification between FTD, AD, mild cognitive impairment (MCI), and controls was attempted, using functional MRI and clinical data, which were analyzed by means of Gradient-Boosted Decision Trees. The accuracy of the model based on functional MRI data achieved a 74% balanced accuracy, with quite good performances on FTD (F1-score = 0.99), but low accuracy in classifying AD, which was mostly misclassified as MCI (F1-score = 0.08). The use of clinical variables as inputs was demonstrated to be appropriate to increase the balanced accuracy to roughly 91%, with a significant increase also in terms of AD classification accuracy (F1-score = 0.74) [227].

The discrimination between FTD and AD was also attempted in underrepresented populations, like the ones of Latin America, by means of ML methods, including SVMs, RF, and other models, reaching 91% accuracy using features like social cognition, neuropsychiatric symptoms, executive functioning performance, and cognitive screening, with minor yet important contributions from age, educational level, and gender [228].

More recently, even more clinically challenging tasks were carried out with the support of ML. Notably, a Canadian group tested the brain patterns eventually associated with FTD variants, with deformation-based morphometry (DBM) proposed to estimate the atrophy of cortical and subcortical areas. Associations between DBM and cognitive test performances were assessed by Partial Least Squares (PLS), followed by linear regression to estimate differences among FTD phenotypes. According to the results, PLS-based atrophy and behavioral patterns were capable of predicting the correct FTD phenotype in 89% of cases, with this percentage slightly dropping to 83% using only brain MRI data and two behavioral tests in the PLS. However, when just using atrophy or behavioral patterns, accuracies fell to 70 and 76%, respectively, highlighting the usefulness of combining MRI and clinical data [229].

Other brain imaging methods, including 2-[18F]fluoro-2-deoxy-D-glucose positron emission tomography, are commonly used in the field of neurodegenerative disorders, especially in those individuals unable or unwilling to undergo MRI scans; however, they have higher impacts in terms of ionizing radiation exposure for the patient and the operator. ML was also applied in this specific investigation, in particular with a multi-class SVM, to classify FTD patients with respect to AD and other conditions. According to the results obtained, the SVM was capable of classifying FTD more accurately than expert readers (78% for pattern-based, 80% for ROI-based classifiers, versus 71% for experts), with the most important features for FTD turning out to be the bilateral frontal cortices and the middle and anterior cingulum [230].

However, the continuous research around the possibility of effectively diagnosing FTD with respect to other neurodegenerative conditions has also been focused on less obtrusive methodologies than MRI, including electroencephalography (EEG) signals, which were investigated in the attempt to identify neuronal and cognitive differences between the two conditions. To this end, six ML approaches were used to classify EEG signals into FTD and AD cases, with the highest performances found with decision trees for AD (78.5% accuracy) and with RF for FTD (86.3% accuracy) [231].

A visual resume of such results is presented in Table 7, and the outline of the most popular AI approaches employed is depicted in Figure 3.

### 3.2. AI for Studying Heavy Metal Exposure’s Effects on Human Health

The research about the effects of heavy metals on human health is quite recent, and this is particularly true when considering those studies where AI has been employed to detect subtle relationships between variables that traditional statistics often overlook.

For example, AI was recently employed to detect the presence of heavy metals in human samples where interference caused by other elements contained within the sample negatively affects the results obtained by inductively coupled plasma mass spectrometry (ICP-MS) approaches, with satisfying results brought by RF, SVMs, Bayesian kernel machine regression (BKMR), extreme gradient boosting (XGBoost), decision trees, multivariate adaptive regression splines, Artificial Neural Networks, boosted trees, and K-nearest neighbors [232,233]. To get more in-depth, the study by Wang and colleagues [233] revealed superior performances by the XGBoost model compared to other AI alternatives in detecting Hg, cobalt (Co), and chromium (Cr) in ICP-MS data, whereas the BKMR algorithm was found to perform better than its counterparts in detecting Pb, Cr, and Co using ICP-MS in whole blood samples, according to Luo and Hendryx [232].

However, focusing on the use of AI in the specific framework of neurological disorders, very recently, the first paper aimed at retrieving evidence of the possible interaction of metal exposure and dietary habits and the worsening of cognitive function was published. The Chinese study used a cohort of older adults and took advantage of AI methods [234]. The research had the remarkable merit of leveraging a popular ML technique, namely the Elastic Net regression model, to identify six potential predictors of cognitive impairment in a cohort of elderly people, including education level, gender, and the frequency of intake of eggs and beans, but also, relevant for our research, the urinary concentration of As and Cd. In particular, MCI risks were boosted by As and Cd concentrations, supporting a role for such metals in the pathophysiology of neurodegenerative conditions.

### 3.3. Future Applications of AI in the Framework of Heavy Metal’s Effects on Neurodegeneration

The recent publication by Liu and colleagues [234] demonstrated the importance of paving the way for new insights into the use of AI in research concerned with heavy metal exposure and neurodegenerative conditions, which can be easily applied to FTD and similar diseases. The availability of new data, often generated with the help of smart (bio)-medical technologies, including Internet-of-Things/Internet-of-Medical-Things (IoT/IoMT) devices [235], also in the framework of neurological disorders, together with the development of new protocols for data analysis based on the higher computational capabilities of current research infrastructures, make it possible nowadays to unravel possibly existing relationships between variables and to eventually shed light on etiopathological and pathophysiological mechanisms yet unknown that could occur in the framework of environmental determinants of neurodegeneration (Figure 4).

## 4. Conclusions

Although it represents the second leading cause of dementia, and despite the vast body of experimental and, in part, epidemiological evidence linking the onset of AD to environmental exposure, the association between toxic metal exposure and the risk of FTD is, to date, almost totally unexplored in both fundamental and clinical research, as is the use of novel methodologies of investigation, including AI. There are myriad biological mechanisms that have been proposed to explain the putative neurotoxicity of certain metals, among which oxidative stress, mitochondrial dysfunction, interference with the release and activities of neurotransmitters, and the hyperphosphorylation of tau protein represent common pathways that act as the major mediators of toxic effects in the brain (Figure 5). In this regard, some valuable information regarding the potential etiopathological role of these non-essential elements in dementia can be retrieved from human studies focused on AD.

Among toxic metals, Al is a known neurotoxin and the only element involved in the formation of NFTs, and epidemiological studies also suggest an association between Al exposure and the risk of dementia, despite a certain heterogeneity of results depending on biological specimens. To the best of our knowledge, Al is currently the only toxic element for which a significant association with EOFTD has been observed. Some epidemiological evidence relying on both community-based and analytical studies using various biological samples has emerged regarding the potential role of As in increasing the risk of AD. However, to date, no studies have been carried out dealing with the association between As exposure and FTD, leaving a relevant topic completely unexplored despite the documented neurotoxic effects of As and the risk associated with elevated exposure to As through diet and drinking water in several areas globally. In addition to the numerous cellular and molecular mechanisms of neurotoxicity associated with Cd, limited epidemiological data support the association between Cd exposure and the risk of AD morbidity and mortality, although some critical issues arise regarding the use of urine to estimate body Cd burden. Furthermore, while studies focused on FTD are needed, the potential of Cd to enter brain tissues should also be definitively established. Although abundant experimental evidence supports the potent neurotoxicity of Pb, and human data have reported a relationship between Pb exposure and cognitive decline, high-quality epidemiological studies directly evaluating the potential ability of Pb to influence the risk of dementia are essentially absent, and investigations conducted to date have shown fragmented and inconsistent results. Although a huge amount of experimental data demonstrate the toxicity of Hg in the CNS, and some results from epidemiological studies are suggestive of a role of Hg in affecting cognitive function, there is currently no solid evidence for a connection between Hg exposure and the risk of AD in longitudinal studies, while research evaluating Hg exposure related to FTD is currently missing.

To summarize, the overall view of the relationships between metal exposure and dementia occurrence or risk is displayed in Figure 6.

Although both are characterized by the anomalous accumulation of tau aggregates in specific brain regions, we should be aware that FTD and AD are two distinct neurodegenerative diseases, with different genetic profiles, molecular drivers, and clinical signs. Therefore, the results of studies evaluating the relationship between exposure to toxic metals and the onset or mortality of AD can provide relevant elements that will necessarily need to be verified and possibly confirmed in subjects suffering from FTD. Unfortunately, despite the heavy impact of dementia on individuals and society, the body of research on the role of toxic metals in the occurrence of dementia and, in general, neurodegenerative disorders is still limited, and very few investigations exploring the levels of toxic metals in large cohorts of AD patients have been published recently. As was already highlighted in the specific sections, the major limitations of the epidemiological studies conducted so far are represented by the small sample sizes, which prevent the production of reliable results, and by the use of ecological or transversal study designs, which are not able to infer a causal directionality in the relationship between metal exposure and disease risk. In addition, considerations around gender are basically absent, and should be investigated in future works to detect eventual differences in the pathophysiology depending on sex/gender. Notably, in retrospective case–control studies, which often measured current exposure rather than assessing past or cumulative exposure (the period in which changes related to AD pathophysiology occur), reverse causality cannot be ruled out. Furthermore, the different biological specimens used to measure metal concentrations are not comparable as they reflect different types of exposure, and it would be desirable to preferentially use hair and nails, biological matrices whose sampling, transport, and storage are easy and feasible; additionally, they can indicate chronic exposure and the actual body burden from metals. Studies using larger sample sizes of patients and considering a number of additional, generally unmeasured, factors (multiple exposures, dietary habits, lifestyle, genetic variants) as confounders in their statistical analyses would provide more reliable estimates of association. Foods, in particular, in addition to being a source of essential and toxic elements, can profoundly influence the incidence of neurodegenerative disorders. Furthermore, analytical studies evaluating the uptake of metals via food and its link with dementia are also warranted. An effective interdisciplinary collaboration between diverse scientist profiles would allow a better understanding of the complex interactions involved in neurodegeneration and the precise cellular and molecular processes involved. Additionally, the use of speciation analysis, biochemical assays, and advanced imaging techniques, recently merged with advanced techniques of AI, particularly dealing with ML models, could help identify reliable biomarkers to assess previous environmental exposure for an early diagnosis and timely treatment.

## Figures and Tables

**Figure 1 antioxidants-13-00938-f001:**
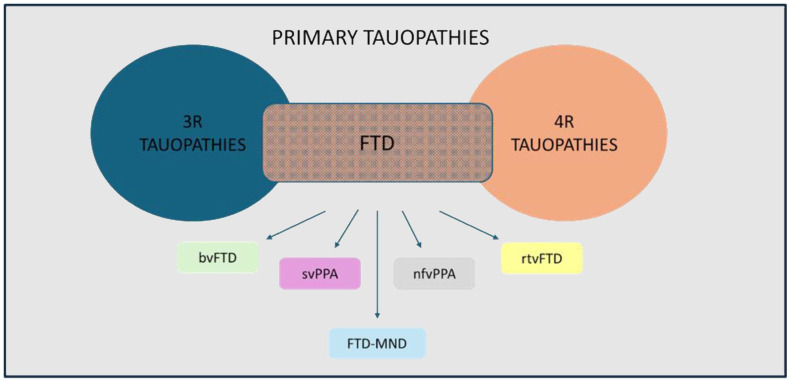
Classification of frontotemporal dementia. Abbreviations: bvFTD: behavioral variant of frontotemporal dementia; FTD: frontotemporal dementia; FTD-MND: frontotemporal dementia associated with motor neuron disease; nfvPPA: nonfluent variant of primary progressive aphasia; rtvFTD: right-lobe variant of frontotemporal dementia; svPPA: semantic variant of primary progressive aphasia.

**Figure 2 antioxidants-13-00938-f002:**
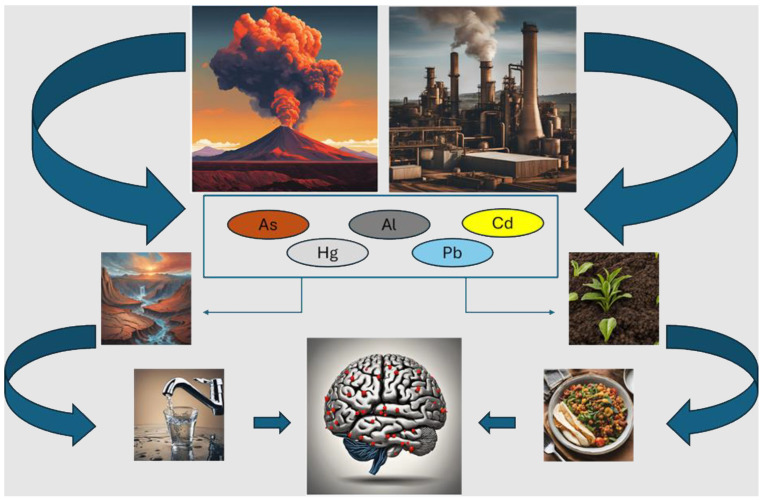
Schematic representation of the sources of contamination and exposure routes of toxic metals. Abbreviations: Al: aluminum; As: arsenic; Cd: cadmium; Hg: mercury; Pb: lead.

**Figure 3 antioxidants-13-00938-f003:**
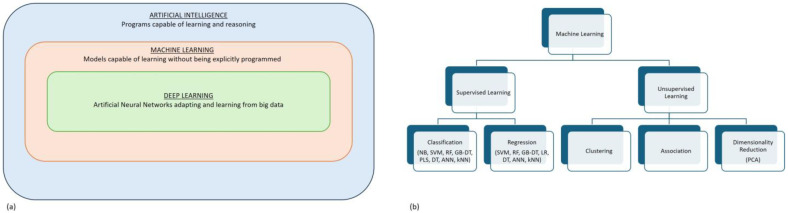
The AI framework: (**a**) the relationship between Artificial Intelligence, machine learning, and Deep Learning; (**b**) the organization tree of machine learning approaches with information about the models used in the investigations for FTD diagnostic purposes (ANN: Artificial Neural Network; DT: decision tree; GB-DT: Gradient-Boosting Decision Tree; kNN: k-nearest neighbors; LR: linear regression; NB: Naïve-Bayes; PCA: Principal Component Analysis; PLS: Partial Least Squares; RF: Random Forest; SVM: Support Vector Machine).

**Figure 4 antioxidants-13-00938-f004:**
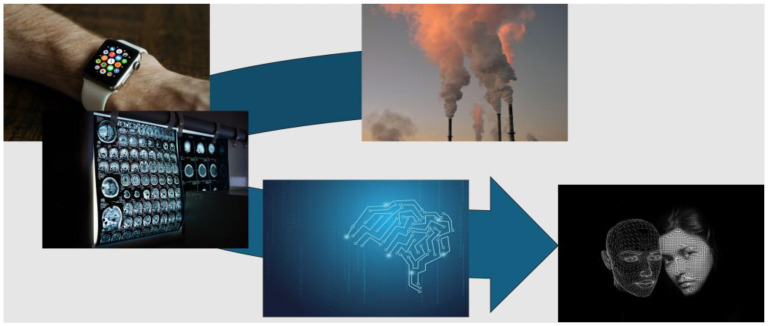
A possible framework for future research in neurodegeneration: heavy metal exposures’ effects can be monitored by means of Internet-of-Things devices and traditional medical diagnostics, whose data, properly analyzed by AI, can drive researchers and clinicians to personalized diagnoses and treatments of neurodegeneration.

**Figure 5 antioxidants-13-00938-f005:**
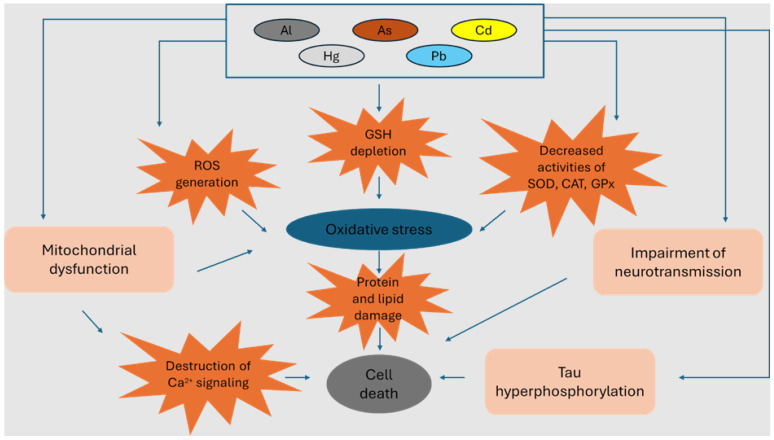
Biological mechanisms underlying neurotoxicity of toxic metals (see text for more detail; Al: aluminum; As: arsenic; Ca: calcium; CAT: catalase; Cd: cadmium; GPx: glutathione peroxidase; GSH: glutathione; Hg: mercury; ROS: reactive oxygen species; SOD: superoxide dismutase).

**Figure 6 antioxidants-13-00938-f006:**
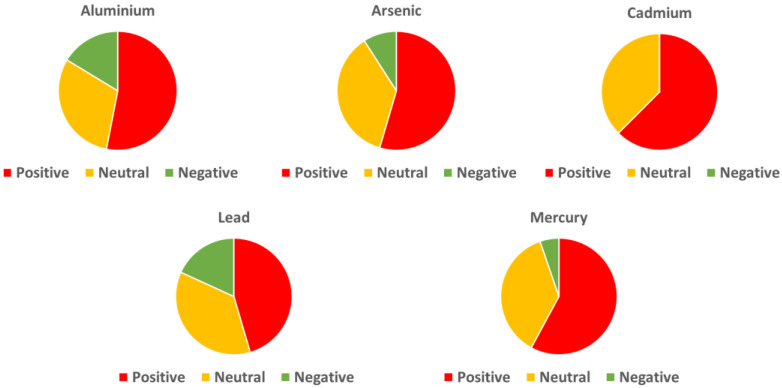
Overview of positive, neutral, and negative relationships, in terms of the number of articles dealing with each related trend, between metal exposure and dementia risks for the different metals considered in the present research.

**Table 1 antioxidants-13-00938-t001:** Primary and secondary tauopathies and their tau isoforms (AD: Alzheimer’s disease; AGD: argyrophilic grain disease; ARTAG: aging-related tau astrogliopathy; CBD: corticobasal degeneration; CTE: chronic traumatic encephalopathy; FTD: frontotemporal dementia; GGT: globular glial tauopathy; PART: primary age-related tauopathy; PD: Parkinson’s disease; PiD: Pick’s disease; PSP: progressive supranuclear palsy).

Condition	Tau Isoform
**Primary tauopathies (tau pathology is predominant)**	
FTD	3R/4R
PSP	4R (predominantly)
PiD	3R (predominantly)
CBD	4R
GGT	4R
AGD	4R
PART	3R/4R
ARTAG	4R
CTE	3R/4R
**Secondary tauopathies (tau pathology is a co-occurring condition)**	
AD	3R/4R
Down syndrome	3R/4R
PD	4R

**Table 2 antioxidants-13-00938-t002:** Clues and pitfalls regarding the association between aluminum and risk of dementia (AD: Alzheimer’s disease; Al: aluminum; CSF: cerebrospinal fluid; EOFTD: early-onset frontotemporal dementia).

Clues	References	Pitfalls	References
Occupational exposure to Al positively associated with the risk of EOFTD	[35]	Conflicting results on the relationship between occupational Al exposure and dementia	[65]
Significantly positive association between Al concentration in drinking water and risk of dementia and AD	[65,66]	Not all routes of exposure considered in the body burden estimate	[66]
Higher mortality rates and higher risk of mortality for AD (but not significant) among miners	[68]	Small sample size and population studies not covering all world areas	[35,71,73,74]
Chronic exposure to Al (drinking water and occupational environment) significantly associated with risk of AD	[69]	Underestimation of mortality for AD in death certificates	[68]
Higher Al levels in serum or blood of AD patients compared to controls	[70,73,74]	Lower levels of Al in serum, hair, and bone in AD subjects than in controls	[70]
Higher serum Al levels in retired workers exposed to Al dust than in controls; levels remained higher for up to 10 years	[71]	Heterogeneity across studies	[70,73]
Inverse association between serum Al levels and cognitive function scores in retired workers	[71]	Publication bias of studies	[73]
Al levels in CSF positively correlated with selected CSF biomarkers for AD	[48]	Limited number of studies included in the meta-analyses	[69,73]
	[66]	Inconsistent results in the comparison between Al concentrations in blood, hair, and CSF	[70]
		No significant correlation between blood Al concentration and upregulation of genes involved in AD pathogenesis	[74]

**Table 3 antioxidants-13-00938-t003:** Clues and pitfalls in the association between arsenic and risk of dementia (AD: Alzheimer’s disease; As: arsenic; CSF: cerebrospinal fluid; iAs: inorganic arsenic; DMA: dimethyl arsenic; MMSE: mini-mental state examination).

Clues	References	Pitfalls	References
Current and long-term exposure to low-level As in groundwater significantly associated with poorer scores in assessments of language, visuospatial skills, and executive functioning	[109]	Lack of direct measurements of environmental As	[109]
Significantly positive correlation of soil As concentration with morbidity and mortality due to AD and other dementias	[110]	Lack of As determination or speciation in individual biological samples	[46,109,110,111,115]
Mortality risk of AD significantly increased with increase in As soil concentration in a dose-dependent fashion	[46]	Complete information on routes/sources of exposure missing	[46,110,116]
High iAs% and low DMA% significantly and positively associated with increased risk of AD	[116]	Lack of adjustment for confounders	[46,48,109,110,111]
Low median level of Se and high median level of iAs%, or/and a low median level of DMA% associated with 2-3-fold significant increased risk of AD	[116]	No significant differences in AD mortality rates between an area with As-contaminated soil and a control area	[111]
Higher iAs% significantly associated with poorer MMSE scores	[116]	Serum As concentration significantly and positively correlated with MMSE scores in AD subjects	[112]
Higher As concentration in hair and nails of AD patients than in controls	[117]	No significant correlation between As concentration and cognitive function scores	[114]
Levels of plasma As positively associated with selected CSF AD biomarkers	[48]	No significant differences in serum/blood As levels between AD patients and controls	[74,112,114]
		Cross-sectional design	[74,112,114,116,117]
		No significant association between urinary As and risk of AD	[115]
		Small sample size	[74,115,117]
		Mild severity of AD patients	[116]
		Other metabolic and/or genetic processes likely involved in the association between As exposure and AD risk	[117]

**Table 4 antioxidants-13-00938-t004:** Clues and pitfalls in the association between cadmium and risk of dementia (AD: Alzheimer’s disease; Cd: cadmium; CSF: cerebrospinal fluid; EOAD: early-onset Alzheimer’s disease; IQR: interquartile range; MMSE: mini-mental state examination).

Clues	References	Pitfalls	References
Significantly higher concentration of Cd in serum/blood in AD subjects than in controls	[73,74]	No significant correlation between occupational exposure to Cd and risk of EOAD	[35]
Significantly positive correlation between blood Cd concentration and risk of AD mortality with a dose–response relationship over a mean-follow-up duration of 5 years	[144]	No significant differences in serum Cd concentration between AD patients and controls	[114]
IQR increase in urinary and blood Cd concentrations significantly associated with higher mortality for AD over a mean follow-up duration of 7.5 years	[146]	Serum Cd levels not significantly correlated with decrease in MMSE scores	[114]
Significantly positive correlation between CSF levels of Cd and targeted AD biomarkers	[48]	No significant correlation between Cd levels and expression of selected genes upregulated in AD	[74]
		Small sample size	[74,114]
		No adjustment for smoking	[48,114]
		Cross-sectional design	[74,114]
		Single blood/urine Cd measurement	[144,146]
		Blood Cd concentration unreliable for past exposure	[144]
		Follow-up period too short	[144]
		Possible underestimation of AD mortality owing to the use of death certificates	[144,146]
		No significant correlation between IQR increase in urinary Cd concentration and AD mortality over a 23-year follow-up and 12.5-year follow-up after adjusting for creatinine	[146]
		Possibility of other risks related to AD mortality beyond Cd exposure	[144,146]
		Heterogeneity between studies	[73]

**Table 5 antioxidants-13-00938-t005:** Clues and pitfalls in the association between lead and risk of dementia (AD: Alzheimer’s disease; CSF: cerebrospinal fluid; EOAD: early-onset Alzheimer’s disease; EOFTD: early-onset frontotemporal dementia; IQR: interquartile range; MMSE: mini-mental state examination; Pb: lead).

Clues	References	Pitfalls	References
Increased levels of Pb in blood and bone significantly associated with worse scores in MMSE in elderly men	[180]	No significant association of occupational Pb exposure with AD, EOAD, and EOFTD	[35,179]
Interaction of patella and blood Pb with age in decreasing MMSE scores in elderly men	[180]	Small sample size	[35]
Higher patella and tibia bone Pb levels significantly associated with cognitive decline over time in elderly men	[181]	Low quality of studies evaluating the relationship between Pb exposure in occupational settings and AD	[179]
Higher patella and tibia Pb concentrations significantly associated with decreased MMSE scores at baseline and over time	[72]	No or inverse association between Pb concentration in biospecimens and AD risk	[70]
Borderline significant association between patella Pb levels and MMSE score < 25	[72]	No significant effect of blood Pb concentration with variations in performance on MMSE test over time	[181]
Positive correlation between CSF Pb concentration and expression levels of selected CSF biomarkers of AD	[48]	No interaction between age and Pb in MMSE score decline	[181]
		No significant association between bone Pb levels and global cognition (both baseline and longitudinal change)	[72]
		Possibility of selection bias due to loss to follow-up	[72]
		Lower circulating Pb levels in AD patients than in control subjects	[73]
		Heterogeneity between studies	[73]
		No significant differences in blood/serum Pb levels between AD patients and healthy controls	[74,182]
		No differences or lower levels of Pb in AD patients than in controls in other biospecimens (CSF, hair, nails, postmortem brains)	[182]
		Possibility of publication bias	[182]
		Lack of adjustment for additional confounders	[48]

**Table 6 antioxidants-13-00938-t006:** Clues and pitfalls in the association between mercury and risk of dementia (AD: Alzheimer’s disease; CSF: cerebrospinal fluid; EOAD: early-onset Alzheimer’s disease; Hg: mercury).

Clues	References	Pitfalls	References
Significant dose–response relationship between urine Hg concentration and cognitive function in workers currently exposed	[204]	Heterogeneity between studies	[73,204]
Some evidence of negative association between past occupational exposure to Hg and current cognitive scores and clinical symptoms	[204]	Only Asian and European subjects considered	[73]
Significantly higher concentration of circulating Hg in AD than in control subjects	[73,74]	Inconsistent results regarding the relationship between Hg levels in various biospecimens and risk of AD, with positive, negative, and lack of association	[70,204]
Levels of plasma Hg positively associated with selected CSF AD biomarkers	[48]	Differences in brain regions assessed and time between autopsy and Hg analysis	[204]
		Decreased nail Hg concentration with increasing AD severity	[205]
		Small sample size	[35,74,204]
		No significant association between occupational Hg exposure and risk of EOAD	[35]
		No correlation between blood Hg concentration and expression levels of selected upregulated genes in AD	[74]

**Table 7 antioxidants-13-00938-t007:** Studies about ML use in FTD (ACC: accuracy; AD: Alzheimer’s disease; ADNI: Alzheimer’s Disease Neuroimaging Initiative; ANN: Artificial Neural Network; bvFTD: behavioral variant of frontotemporal dementia; DLB: dementia with Lewy bodies; DT: decision tree; EEG: Electroencephalography; FDG-PET: fluorodeoxyglucose positron emission tomography; FTD: frontotemporal dementia; FTLDNI: Frontotemporal Lobar Degeneration Neuroimaging Initiative; GB-DT: Gradient-Boosting Decision Tree; HC: healthy controls; kNN: k-nearest neighbors; LR: linear regression; MBACC: mean balanced accuracy; MCI: mild cognitive impairment; ML: machine learning; MMAAUC: mean macro-averaged AUC; MMAF1: mean macro-averaged F1-score; MRI: magnetic resonance imaging; NB: Naïve-Bayes; nfvPPA: nonfluent variant primary progressive aphasia; PCA: Principal Component Analysis; PLS: Partial Least Squares; RF: Random Forest; rs-fMRI: resting state functional magnetic resonance imaging; SE: sensitivity; SP: specificity; SVM: Support Vector Machine; svPPA: semantic variant primary progressive aphasia).

Population	Methods	ML Approach	Results	References
170 AD, 72 bvFTD, 87 HC	Neuropsychological tests	NB, SVM	>84% accuracy in diagnosis and differential diagnosis.	[223]
153 AD, 87 FTD, 99 HC	MRI data	PCA, SVM	Cross-sectional study: differentiation of AD vs. HC at 83.3%, FTD vs. HC at 82.1%, FTD vs. AD at 63.3%. Longitudinal study: differentiation of AD vs. HC at 90.0%, FTD vs. HC at 88.0%, FTD vs. AD at 75.0%.	[225]
515 bvFTD divided into independent training and validation cohorts	MRI data and semantic fluency test	RF	Validation cohort. MRI only: ACC: 88%, SE: 81%, SP: 92%. MRI and semantic fluency: ACC: 91%, SE: 79%, SP: 96%.	[226]
32 AD, 151 FTD, 103 MCI, 51 HC from ADNI, 96 HC from FTLDNI	rs-fMRI and clinical variables	GB-DT	MRI only: MBACC: 74.4%, MMAAUC: 0.94, MMAF1: 0.73. F1: FTD: 0.99, HC: 0.99, MCI: 0.86, AD: 0.08. MRI and clinical: MBACC: 91.1%, MMAAUC: 0.99, MMAF1: 0.92. F1 AD: 0.74.	[227]
904 AD, 282 FTD, 606 HC	Demographic, clinical, and cognitive data	RF	AD vs. FTD: accuracy = 0.91.	[228]
70 bvFTD, 36 svPPA, 30 nfvPPA	Brain MRI, clinical and neuropsychological examination	PLS and LR	MRI and behavioral pattern: ACC: 89.12%, SP: 91.46–97.15%, SE: 84.19–93.56%. MRI and two behavioral tests: ACC: 83.62%, SP: 86.38–93.51%, SE: 76.17–87.50%. MRI: ACC: 69.76%. Behavior: ACC: 76.38%.	[229]
63 AD, 79 DLB, 23 FTD, 41 HC	FDG-PET	SVM	F1-scores: AD: 0.74, DLB: 0.81, FTD: 0.87, HC: 0.71.	[230]
10 AD, 10 FTD, 8 HC	EEG	DT, RF, ANN, SVM, NB, kNN	Best ACC: AD: 78.5% (DT), FTD: 86.3% (RF).	[231]

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
