# Peer review of "Metal Toxicity and Dementia Including Frontotemporal Dementia: Current State of Knowledge"

_antioxidants, 2024, doi:10.3390/antiox13080938_

Round 1

Reviewer 1 Report

The authors provide an extraordinarily comprehensive literature review on the impact of metal toxicity on FTD and related dementias.  

The exploration of clues, evidence, and pitfalls in Table format for each of the major metals was a key strength of the paper.

The authors do report quantitative effect sizes for several studies. However, they are buried among the paragraphs of text. While I understand this is NOT a meta-analysis, it would still greatly enhance the paper if the authors could compile the effect sizes they have already extracted into aggregate statistic that better estimates the overall effect size of metal toxicity on dementia burden - albeit incidence or severity.

The AI section was a nice addition, but the figure for the AI section is trivial and difficult to make out.  I'm not sure this figure in its current states adds anything to a paper that is already so large.

Again, the comprehensiveness of the review is appreciated. However, 45 pages in a MDPI journal for a review without other quantitative analytical contributions is very, very large.  It would improve the readability of the paper if formatting (bullets, numbers, more sub-section, etc.) could be used to reduce long paragraphs, long sentences, and unnecessarily verbose transitory text.

See major comments above.

Author Response

We are grateful to the reviewer for their constructive comments. We provide here a point-to-point response (in italics) to their concerns (in plain text), hoping having been able to properly respond to the points they have raised.

Reviewer 1

The authors provide an extraordinarily comprehensive literature review on the impact of metal toxicity on FTD and related dementias. 

The exploration of clues, evidence, and pitfalls in Table format for each of the major metals was a key strength of the paper.

We thank the reviewer for the appreciation of our work.

The authors do report quantitative effect sizes for several studies. However, they are buried among the paragraphs of text. While I understand this is NOT a meta-analysis, it would still greatly enhance the paper if the authors could compile the effect sizes they have already extracted into aggregate statistic that better estimates the overall effect size of metal toxicity on dementia burden - albeit incidence or severity.

Thank you very much for this comment. We perfectly understand that an effect size measurement of the impact of metal exposure on the risk of dementia/AD could enrich the text, however, due to the general paucity of published studies and to great heterogeneity of endpoints (incidence, mortality) evaluated and, consequently, different risk measures (relative risk, hazard ratio, odds ratio), the production of an aggregate statistic appears quite difficult and, eventually, scattered, possibly leading to spurious considerations. However, to the best of our knowledge, this is the first comprehensive review analysing the epidemiological evidence and underlying molecular mechanisms on the relationship between toxic metal exposure and dementia risk, we therefore hope that it can also provide valuable information and indications for future studies on this topic which has not yet been sufficiently explored. In any case, with Figure 6, we tried providing an overview, at a glance, of possible contributions to dementia by the metals considered. In particular, Figure 6 includes, for each of the metals object of the present investigation, the proportion of articles reporting positive, neutral or negative relationships, between exposure and dementia risk or development. We hope this figure could be helpful to the reader to better identify the main contributors in this regard at a glance.

The AI section was a nice addition, but the figure for the AI section is trivial and difficult to make out.  I'm not sure this figure in its current states adds anything to a paper that is already so large.

Thank You for Your kind words and encouragement. We modified Figure 3, postponing it in the main text and providing more information regarding the actual use of AI models in the framework of FTD research, highlighting the sub-categories they belong to, for a better understanding by the reader and in order to provide the reader with more relevant information.

Again, the comprehensiveness of the review is appreciated. However, 45 pages in a MDPI journal for a review without other quantitative analytical contributions is very, very large.  It would improve the readability of the paper if formatting (bullets, numbers, more sub-section, etc.) could be used to reduce long paragraphs, long sentences, and unnecessarily verbose transitory text.

Thank you for the valuable comment. We entirely revised the text, with the aim to improve readability and, where possible, to reduce verbosity of sentences.

Reviewer 2 Report

The manuscript "Metal Toxicity and Frontotemporal Dementia: What Do We Know So Far?" is a significant advancement in the field, written by experts. It focuses on the critical topic of heavy metal activity in relation to dementia, presenting important information despite the topic not being recent.The introduction of the topic of artificial intelligence and its applications in this field certainly makes it more recent.

Here are my observations:

The manuscript is complete in its parts but  where appropriate, but authors are encouraged to include a section addressing sex and gender considerations where appropriate.

Please ensure to carefully review the sections of the manuscript that have been highlighted in the iThenticate report.

With these additions made the manuscript can be accepted.

Author Response

We are grateful to the reviewer for their constructive comments. We provide here a point-to-point response (in italics) to their concerns (in plain text), hoping having been able to properly respond to the points they have raised.

Reviewer 2

Major comments

The manuscript "Metal Toxicity and Frontotemporal Dementia: What Do We Know So Far?" is a significant advancement in the field, written by experts. It focuses on the critical topic of heavy metal activity in relation to dementia, presenting important information despite the topic not being recent. The introduction of the topic of artificial intelligence and its applications in this field certainly makes it more recent.

Thank You very much for Your kind comment.

Here are my observations:

Detail comments

The manuscript is complete in its parts but  where appropriate, but authors are encouraged to include a section addressing sex and gender considerations where appropriate.

Thank You for Your kind observation. The main problem with this is that the articles included, which are already quite heterogeneous, do not face gender analysis, making our further investigation in this sense, to the best, speculative. However, we included a limitation acknowledgment in this sense in the Conclusions section.

Please ensure to carefully review the sections of the manuscript that have been highlighted in the iThenticate report.

Thank You. We performed modifications throughout the manuscript accordingly.

With these additions made the manuscript can be accepted.

Thank You, much appreciated.

Reviewer 3 Report

Overall, this is a thoughtful and timely review that addresses a very understudied topic-the role of metal toxicity in FTD. The authors generally cite up to date references and the tables are very useful. I especially appreciated the tables that lay out the evidence for and against the role of different metals in FTD and, mostly, AD. I do have two major concerns and also a number of points that need to be clarified as listed below. My first major concern is the title. Most of the data cited regarding metal toxicity is for AD not FTD. Thus, I feel that the title is misleading and should be changed to something such as “Metal toxicity and Dementia including FTD: Current state of knowledge”. The second major concern is the sections on underlying mechanisms. In contrast to the discussions of the epidemiological data, the discussions of mechanisms lack any caveats or critical assessments. The authors need to be more critical in their discussion of these mechanisms, especially as in some cases they seem to be contradictory. Also, some of the same mechanisms are listed more than once for the same metal.

Points to address/clarify:

1.        Line 29: what is meant by “sanitary perspectives”?

2.        Table 1 needs a legend that explains the different diseases. Also, not all of the primary tauopathies and only a subset of the secondary tauopathies listed in the text are included in the table. Please justify the exclusion of these diseases from the Table or include them.

3.        Lines 136-137: Earlier AGD, CBD, PiD, etc are described as separate diseases from FTD. Why are they called FTD-Tau subtypes here? This is very confusing for the reader who is not highly familiar with the field.

4.        Lines 284-287: This sentence is quite confusing. Is a fetus considered one of the tissues that Al can distribute to? Please re-write to clarify this point. Also, if Al mainly accumulates in bone does that mean that it is slowly released over time to other tissues as described for some of the other metals?

5.        Lines 335-337: This sentence makes no sense. Do you mean “consuming” rather than “assuming”?

6.        Table 2, Pitfalls, ref 73: “by” is not needed here. This should be changed to “publication bias of studies”.

7.        Lines 450-454: It is not clear why the binding of Al to ATP specifically reduces the sensitivity of Ca-dependent inactivation of NMDA receptors. Please explain further.

8.        Section 2: Please be consistent in your naming of the different forms of arsenic in this section. There is currently quite a lot of variation. Also, when As alone is used in the text, what form does that refer to? Please clarify.

9.        Lines 507-508: What is meant by “a broader range of domains than that associated with the current exposure”?

10.  Lines 525-530: The description of this paper is not clear. The authors first say that serum levels of As were strongly correlated with increased MMSE score and suggest it is due to higher arsenic levels reflecting higher fish consumption. However, they then say that there was no difference in serum As concentration between controls and AD patients. If As levels correlate with MMSE score and, by definition, AD patients should have lower MMSE scores than controls, then they should have lower serum As levels. Please explain.

11.  Lines 539-540: The authors state here that urinary As accounts for recent As exposure while AD is a chronic disease. However, couldn’t the same argument be made about serum levels of As? Please clarify.

12.  Line 570: What is the ecological fallacy?

13.  Lines 592-593: A word is missing between “major” and “responsible”.

14.  Lines 593-594: Why would inhibition of succinate dehydrogenase activity impair the function of Complex I and uncouple oxidative phosphorylation? Succinate dehydrogenase is Complex II and is not part of coupling the ETC to ATP generation.

15.  Lines 611-616: Inhibition of the Akt pathway and activation of the AMPK pathway is associated with longevity and maintenance of cognitive function so I am not sure that these effects of As would actually be associated with toxicity.

16.  Line 632: nuclear factor E2-related factor is Nrf2 and its activation is generally considered beneficial in the context of oxidative stress and neurodegeneration. Thus, here too, I don’t see why its activation would contribute to As toxicity.

17.  Lines 699-701: Here the authors say that the level of Cd in the urine is reflective of the real body burden which is the opposite of what they said about As (eg lines 539-540). Please explain the differences between the body’s handling of the two metals that makes urine useful for studying Cd burden but not As burden.

18.  Lines 718-720: the portion of the sentence beginning with “however” does not make sense. Please clarify.

19.  Lines 728-732: Why is this study on Al included in the Cd section?

20.  Lines 790-795: I do not understand why an upregulation of proteins required for autophagy along with an increase in autophagosomes would inhibit autophagy. It seems to me that it would increase autophagy.

21.  Lines 844-856: This sentence is not clear. Are these two different sources of Pb exposure?

22.  Line 847: Do the authors mean “up to” or “more than” here?

23.  Line 888: Is each quartile 20 µg/g? Please clarify.

24.  Line 902: Please define IQR and explain this finding more clearly.

25.  Lines 907-909: This study seems to contradict the study described in lines 897-901. Is that correct?

26.  Lines 981: Please explain why an increase in intracellular calcium would enhance the ETC chain.

27.  The relevance of section 3.1 to the overall review is not clear. The discussion of how to use AI to interrogate further the effects of metals on human health is interesting but its use for diagnosing FTD is not directly relevant to this review. I suggest removing section 3.1, especially as the review is already quite long.  

28.  Line 1334: I would say “supporting” rather than “confirming” here. Also, why did the authors switch from metals to heavy metals ,especially as earlier they describe As as a metalloid.

29.  Figure 4 legend: Define IoT.

30.  Line 1373: I think “EAFTD” should be “EOFTD”.

31.  Line 1406: Explain what ecological and transversal study designs are.

Author Response

We are grateful to the reviewer for their constructive comments. We provide here a point-to-point response (in italics) to their concerns (in plain text), hoping having been able to properly respond to the points they have raised.

Reviewer 3

Overall, this is a thoughtful and timely review that addresses a very understudied topic-the role of metal toxicity in FTD. The authors generally cite up to date references and the tables are very useful. I especially appreciated the tables that lay out the evidence for and against the role of different metals in FTD and, mostly, AD.

We thank the reviewer for the overall positive evaluation of our work.

I do have two major concerns and also a number of points that need to be clarified as listed below. My first major concern is the title. Most of the data cited regarding metal toxicity is for AD not FTD. Thus, I feel that the title is misleading and should be changed to something such as “Metal toxicity and Dementia including FTD: Current state of knowledge”.

Thank you for this valuable comment. We have changed the title as you suggested.

The second major concern is the sections on underlying mechanisms. In contrast to the discussions of the epidemiological data, the discussions of mechanisms lack any caveats or critical assessments. The authors need to be more critical in their discussion of these mechanisms, especially as in some cases they seem to be contradictory. Also, some of the same mechanisms are listed more than once for the same metal.

Thank you for this comment. As specified at the beginning of the specific sections, the molecular mechanisms that explain the neurotoxicity of metals are not completely clarified in the literature itself, therefore, in order to reduce assumptions that could have been regarded as mainly speculative, we attempted at summarizing the proposed processes based on the main evidence data from in vitro and in vivo studies. Given the great heterogeneity and uncertainty of the mechanisms, since an analytical discussion is quite difficult and would surely carry on biases and speculation, we provided a schematic representation of the common processes through which toxic metals can exert neurotoxic effects. However, we carefully revised the text and delete useless repetitions.

Points to address/clarify:

  1. Line 29: what is meant by “sanitary perspectives”?

We clarified accordingly.

  1. Table 1 needs a legend that explains the different diseases. Also, not all of the primary tauopathies and only a subset of the secondary tauopathies listed in the text are included in the table. Please justify the exclusion of these diseases from the Table or include them.

Thank You, our mistakes. All diseases are now in the Table accordingly. Legend explaining the acronyms is present at the table caption (above the table).

  1. Lines 136-137: Earlier AGD, CBD, PiD, etc are described as separate diseases from FTD. Why are they called FTD-Tau subtypes here? This is very confusing for the reader who is not highly familiar with the field.

Thank You. We removed part of the sentence to avoid creating confusion in the less familiar readers.

  1. Lines 284-287: This sentence is quite confusing. Is a fetus considered one of the tissues that Al can distribute to? Please re-write to clarify this point. Also, if Al mainly accumulates in bone does that mean that it is slowly released over time to other tissues as described for some of the other metals?

We apologize for this error. We rephrased the sentence. After absorption, Al is distributed unequally in human tissues and accumulates in some of them. Half of the body's burden of Al is found in the skeleton where it can cause fractures accompanied by osteomalacia, thus bone Al appears to produce toxic effects locally and cannot be considered as a pool from which it is released into the circulation over time as occurs with lead.

  1. Lines 335-337: This sentence makes no sense. Do you mean “consuming” rather than “assuming”?

We amended the typo. Thank you.

  1. Table 2, Pitfalls, ref 73: “by” is not needed here. This should be changed to “publication bias of studies”.

Thanks for this comment. We corrected this grammar error accordingly.

  1. Lines 450-454: It is not clear why the binding of Al to ATP specifically reduces the sensitivity of Ca-dependent inactivation of NMDA receptors. Please explain further.

Thank you for the valuable comment. We added a detailed explanation of this mechanism.

  1. Section 2: Please be consistent in your naming of the different forms of arsenic in this section. There is currently quite a lot of variation. Also, when As alone is used in the text, what form does that refer to? Please clarify.

Thank you for this comment. When not specified, As alone refers to the total As, and not to a single species.

  1. Lines 507-508: What is meant by “a broader range of domains than that associated with the current exposure”?

Thank You for pointing out. We clarified the sentence.

  1. Lines 525-530: The description of this paper is not clear. The authors first say that serum levels of As were strongly correlated with increased MMSE score and suggest it is due to higher arsenic levels reflecting higher fish consumption. However, they then say that there was no difference in serum As concentration between controls and AD patients. If As levels correlate with MMSE score and, by definition, AD patients should have lower MMSE scores than controls, then they should have lower serum As levels. Please explain.

Thank You for pointing that out, allowing us to better explain the point. Serum As was not different between AD and control subjects; however, the correlation between MMSE and serum As levels (significant even after the Bonferroni correction) was observed among individuals with AD, since MMSE was performed only among AD subjects. We have now specified that in the text. Additionally, seafood mainly contains non-toxic organic species of As: we also added this point to the text.

  1. Lines 539-540: The authors state here that urinary As accounts for recent As exposure while AD is a chronic disease. However, couldn’t the same argument be made about serum levels of As? Please clarify.

Thank you for this important comment. You are right. We deleted this sentence and implemented the final summary of this section with this statement: “Conversely, the analytical studies have provided conflicting results, especially if based on As measurements in urinary samples, which, similarly to As content in serum specimens, reflect only recent exposure, unlike AD, which is a chronic disorder. Therefore, large longitudinal cohort studies should be conducted differentiating As species in biospecimens such as nails to assess both the past exposures and the link between As metabolism capability and risk of dementia and collecting information on a wide range of confounding factors, particularly regarding genetic analysis, diet and potential sources of exposure.

  1. Line 570: What is the ecological fallacy?

Ecological fallacy is commonly attributed to community-based studies, where it is assumed that results dealing with the characteristics of a group are attributed to its single individuals. For example, when economic studies are carried out, a given region can be considered as wealthier to another, just because of the presence of a few extremely rich inhabitants.

  1. Lines 592-593: A word is missing between “major” and “responsible”.

Thank You, corrected.

  1. Lines 593-594: Why would inhibition of succinate dehydrogenase activity impair the function of Complex I and uncouple oxidative phosphorylation? Succinate dehydrogenase is Complex II and is not part of coupling the ETC to ATP generation.

We apologize for this imprecision. We corrected accordingly.

  1. Lines 611-616: Inhibition of the Akt pathway and activation of the AMPK pathway is associated with longevity and maintenance of cognitive function so I am not sure that these effects of As would actually be associated with toxicity.

Thank you for raising this interesting point. Actually, Akt activity and Akt levels are decreased in the brains of AD and PD patients, linking Akt signal and neurodegenerative development (reviewed in doi: 10.1016/j.neuro.2021.05.00). Over-expression of AMPK signaling has been detected in several brain diseases, including neurodegenerative diseases, suggesting a correlation between AMPK activation and NDs (doi: 10.1007/978-3-319-43589-3_7). In addition, a number of studies reported that decreased Akt phosphorylation and/or the activation of AMPK signal contribute to neuronal cell apoptosis by exposure to chemicals (doi: 10.1016/j.tox.2019.152245; 10.1016/j.cellsig.2014.04.009; 10.1186/s13041-016-0194-6).

  1. Line 632: nuclear factor E2-related factor is Nrf2 and its activation is generally considered beneficial in the context of oxidative stress and neurodegeneration. Thus, here too, I don’t see why its activation would contribute to As toxicity.

Thank you for this relevant comment. Our mistake. We deleted this sentence.

  1. Lines 699-701: Here the authors say that the level of Cd in the urine is reflective of the real body burden which is the opposite of what they said about As (eg lines 539-540). Please explain the differences between the body’s handling of the two metals that makes urine useful for studying Cd burden but not As burden.

This is an interesting point, thank you. As stated, “the kidneys are the major organs in which Cd accumulates (Cd half-life between 6-38 years), therefore the concentration of Cd in the urine is suggestive of chronic exposure”. In contrast, urinary As, although, is the sample of choice for As analysis as As is excreted predominantly by the kidney, has a biological half-life of approximately 4 days, thus it reflects the recent exposure.

  1. Lines 718-720: the portion of the sentence beginning with “however” does not make sense. Please clarify.

Thank you for this comment. Adjustment for urinary creatinine renders the association between urinary Cd exposure and AD mortality no longer significant. This demonstrates that creatinine is a confounding factor to be considered in risk associations.

  1. Lines 728-732: Why is this study on Al included in the Cd section?

We apologize for this error. This meta-analysis evaluated the circulating levels of several toxic metals (aluminum, mercury, cadmium, lead) in AD patients and controls. Data in the text refers to Cd, and we corrected accordingly.

  1. Lines 790-795: I do not understand why an upregulation of proteins required for autophagy along with an increase in autophagosomes would inhibit autophagy. It seems to me that it would increase autophagy.

Thank you for your reliable observation. We corrected the sentence (lines 862-866).

  1. Lines 844-856: This sentence is not clear. Are these two different sources of Pb exposure?

We apologize for this imprecision. Exposure to lead may occur via inhalation and ingestion, but the proportion of Pb absorbed after ingestion may vary by fasting conditions.

  1. Line 847: Do the authors mean “up to” or “more than” here?

We corrected this sentence, thank you.

  1. Line 888: Is each quartile 20 µg/g? Please clarify.

Thank you for this comment. Each increase in 20 µg Pb/g patella bone (corresponding to an interquartile range) is associated with a decreased MMSE score. We clarified further this point in the text (line 962),

  1. Line 902: Please define IQR and explain this finding more clearly.

Thank you for the valuable comment. The full name of IQR was provided on line 708. We better clarified this point in the text.

  1. Lines 907-909: This study seems to contradict the study described in lines 897-901. Is that correct?

Thank you for this relevant observation. The study is the same, and evaluated both MMSE, which evaluates several domains including memory, visuospatial ability, attention, language, and orientation, as well as global cognition, which is the sum of scores obtained from other 7 individual tests in used in the NES2 (Neurobehavioral Evaluation System 2), CERAD (Consortium to Establish a Registry for Alzheimer’s Disease), or WAIS-R (Wechsler Adult Intelligence Scale-Revised). We added a brief explanation to the text (line 983).

  1. Lines 981: Please explain why an increase in intracellular calcium would enhance the ETC chain.

Thank you for this comment. Ca2+ is believed to regulate mitochondrial oxidative phosphorylation. Indeed, it can directly induce ATP production through activation of the F1Fo-ATP synthase and stimulate flux through Complex III of the electron transport chain (doi: 10.1021/bi3015983). In addition, Ca2+ plays a major role in the regulation of mitochondrial function by boosting the production of NADH and FADH2 through enhancements in the activities of several mitochondrial substrate transporters and dehydrogenase enzymes, thereby enhancing mitochondrial respiratory chain activity with a subsequent increase in H+ pumping, proton motive force generation, O2 consumption, and ATP synthesis (doi: 10.3390/cells11010131).

  1. The relevance of section 3.1 to the overall review is not clear. The discussion of how to use AI to interrogate further the effects of metals on human health is interesting but its use for diagnosing FTD is not directly relevant to this review. I suggest removing section 3.1, especially as the review is already quite long.

Thank You for Your kind suggestion. In fact, we decided to keep Section 3 as the two other reviewers have highlighted its added value with respect to the current literature and in the perspective of the future developments in the field. However, we agree with you about the length of the paper, overall, therefore we tried to reduce redundancy and verbosity throughout the manuscript to keep it readable to the audience as much as possible.

  1. Line 1334: I would say “supporting” rather than “confirming” here. Also, why did the authors switch from metals to heavy metals ,especially as earlier they describe As as a metalloid.

Thank You, our mistake. We changed accordingly.

  1. Figure 4 legend: Define IoT.

Done.

  1. Line 1373: I think “EAFTD” should be “EOFTD”.

Right. Changed accordingly. Thank You!

  1. Line 1406: Explain what ecological and transversal study designs are.

Thank you for giving us the possibility to clarify these key concepts. Studies with ecological design are descriptive design, thus describe and analyze correlations between different variables. Although the unit of analysis is aggregated data from multiple individuals, ecological studies allow to investigate large populations.  Cross-sectional or transversal studies represent the basic design of analytical studies, since they simultaneously collect individual-level variables; therefore, although they are useful for estimating prevalence, they are not able, like ecological studies, to establish causal relationships.